# Spontaneous human CD8 T cell and autoimmune encephalomyelitis-induced CD4/CD8 T cell lesions in the brain and spinal cord of HLA-DRB1*15-positive multiple sclerosis humanized immune system mice

Irini Papazian[1], Maria Kourouvani[1,2], Anastasia Dagkonaki[1], Vasileios Gouzouasis[1,3], Lila Dimitrakopoulou[4], Nikolaos Markoglou[5], Fotis Badounas[1,6], Theodore Tselios[7], Maria Anagnostouli[5], Lesley Probert[1]*

[1]Laboratory of Molecular Genetics, Hellenic Pasteur Institute, Athens, Greece; [2]Athens International Master's Programme in Neurosciences, Department of Biology, National and Kapodistrian University of Athens, Athens, Greece; [3]Department of Molecular Biology and Genetics, Democritus University of Thrace, Alexandroupolis, Greece; [4]Department of Hematology, Laiko General Hospital, National and Kapodistrian University of Athens, Athens, Greece; [5]Research Immunogenetics Laboratory, Multiple Sclerosis and Demyelinating Diseases Unit, First Department of Neurology, School of Medicine, National and Kapodistrian University of Athens, NKUA, Aeginition University Hospital, Athens, Greece; [6]Transgenic Technology Unit, Hellenic Pasteur Institute, Athens, Greece; [7]Department of Chemistry, University of Patras, Patras, Greece

*For correspondence:
lesley.probert@gmail.com

Competing interest: The authors declare that no competing interests exist.

**Abstract** Autoimmune diseases of the central nervous system (CNS) such as multiple sclerosis (MS) are only partially represented in current experimental models and the development of humanized immune mice is crucial for better understanding of immunopathogenesis and testing of therapeutics. We describe a humanized mouse model with several key features of MS. Severely immunodeficient B2m-NOG mice were transplanted with peripheral blood mononuclear cells (PBMCs) from HLA-DRB1-typed MS and healthy (HI) donors and showed rapid engraftment by human T and B lymphocytes. Mice receiving cells from MS patients with recent/ongoing Epstein–Barr virus reactivation showed high B cell engraftment capacity. Both HLA-DRB1*15 (DR15) MS and DR15 HI mice, not HLA-DRB1*13 MS mice, developed human T cell infiltration of CNS borders and parenchyma. DR15 MS mice uniquely developed inflammatory lesions in brain and spinal cord gray matter, with spontaneous, hCD8 T cell lesions, and mixed hCD8/hCD4 T cell lesions in EAE immunized mice, with variation in localization and severity between different patient donors. Main limitations of this model for further development are poor monocyte engraftment and lack of demyelination, lymph node organization, and IgG responses. These results show that PBMC humanized mice represent promising research tools for investigating MS immunopathology in a patient-specific approach.

## eLife assessment

The humanized immune system model represents a **valuable** model in which to evaluate mechanisms that may drive MS processes in vivo. The data are **solid** given the revisions and expansion of numbers of mice to yield more statistical rigor. This model will be used by the greater community studying MS pathophysiology.

## Introduction

Multiple sclerosis (MS) is a complex chronic inflammatory demyelinating and neurodegenerative disease of the central nervous system (CNS) with autoimmune characteristics. Pathological hallmarks are focal plaques of immune-mediated demyelination in the white matter, inflammation at CNS borders and diffuse demyelination and neurodegeneration of the gray and white matter of the brain and spinal cord. Genome-wide association studies show that the vast majority of MS risk genes are related to immune activation and function, thereby strongly supporting the role of adaptive immunity in MS pathogenesis. These include the strongest MS-associated genetic risk factor, the human leukocyte antigen (HLA) DRB1*15, specifically the HLA-DRB1*15:01 variant, of the major histocompatibility complex (MHC) class II that restricts CD4+ T cells (*Sawcer et al., 2011*; *Beecham et al., 2013*). Pathological analyses of autopsy tissues reveal CD8+ lymphocytes and monocytes as the main CNS infiltrating cells, with B cells accumulating in meningeal and perivascular spaces, and relatively few infiltrating CD4+ T cells (*Lassmann, 2019*). Studies in animal models, mainly experimental autoimmune encephalomyelitis (EAE), showing that myelin-reactive T cells together with peripheral monocytes are necessary for EAE induction, as well as the effectiveness of current immunotherapies in MS patients that work mainly by targeting peripheral immune responses, further support an autoimmune pathogenesis for MS.

EAE and MS share many clinical, immunological, and pathological similarities (*Regen and Waisman, 2021*; *Schreiner et al., 2009*), but they also present significant differences (*Attfield et al., 2012*; *Lassmann and Bradl, 2017*). One difference is the predominance of CD8+ T cells in MS lesions, while these cells are scarce in the CNS of EAE mice (*Babbe et al., 2000*). In addition, the inflammatory process in EAE primarily targets the spinal cord, whereas in MS it also targets the brain. Moreover, while some MS drugs were discovered from research in the EAE model (natalizumab, glatiramer acetate), there is a relatively low or moderate success rate of drugs that showed high efficacy in both the EAE model and the human disease (*Wiendl and Hohlfeld, 2009*). These differences infer that different pathways or mechanisms are prevalent in MS versus EAE (*Wekerle et al., 2012*; *Mestas and Hughes, 2004*). Differences are further highlighted by recent clinical data showing the critical importance of B cells to MS pathology, a cell population that does not show important functional participation in most EAE models (*Hauser et al., 2017*; *Li and Bar-Or, 2019*). The development of animal models capable of developing a human immune system is crucial to the study of human immune-mediated diseases such as MS, and for evaluating the safety and efficacy of human immunotherapies. Most current studies with humanized mouse models use immunodeficient non-obese diabetic (NOD) – severe combined immunodeficiency disorder (*scid*) mice deficient or mutant for the common cytokine receptor γ-chain gene (*IL2rg*) (named NSG or NOG mice, respectively) that lack murine B, T cells, and NK cells and have defective innate immune responses (*Allen et al., 2019*; *Nagatani et al., 2019*; *Willinger et al., 2011*; *Morillon et al., 2020a*). These mouse strains readily engraft with human peripheral blood mononuclear cells (PBMCs) without prior irradiation, as murine MHC molecules support the proliferation of both human CD4+ and CD8+ T cells, with CD8+ T cells predominating. Recent variants of these strains lacking MHC class I molecules (B2m) show increased CD4 to CD8 ratios and a longer time window before onset of xenogeneic graft versus host disease (GVHD) and represent improved strains for the study of human immune responses (*King et al., 2009a*; *Morillon et al., 2020b*). Previous studies showed that PBMC humanized NSG mice with cells from healthy donors are susceptible to EAE, exhibiting subclinical disease, with infiltration of the brain and spinal cord by CD4+ and CD8+ T cells, but not monocytes (*Zayoud et al., 2013*).

To investigate the potential of PBMC from MS patients to induce CNS-directed immunopathology in humanized mice, we engrafted B2m-NOD/*scid*-IL2Rγ$^{null}$ (B2m-NOG) mice with PBMC from human donors based on HLA-DRB1 genotype, relapsing remitting MS (RRMS) diagnosis and therapy, and monitored CNS inflammation that developed spontaneously or after immunization for EAE with

myelin antigens. We show that PBMC from MS and healthy (HI) donors rapidly engraft B2m-NOG mice with human (h) CD4$^+$ and hCD8$^+$ T and B cells, with donor-specific differences toward proportions of human immune cell populations engrafted and propensity to develop CNS inflammation. Mice transplanted with PBMC from HLA-DRB1*15-positive (DR15) MS patients uniquely developed spontaneous hCD8$^+$ T cell lesions in the spinal cord gray matter and brainstem, and prominent brain white and gray matter lesions with mixed hCD4/hCD8 T cells in mice immunized for EAE with myelin peptides. Our results show that PBMC humanized B2m-NOG mice partially reproduce MS immunopathology and can be used to investigate human immune responses toward CNS in a simple, rapid, and personalized manner.

## Results

### Reconstitution of human adaptive immune system in B2m-NOG mice engrafted with PBMC from MS patient and HI donors

Six long-term RRMS patients, of which five were HLA-DRB1*15-positive (DR15 MS1-5) and one HLA-DRB1*13-positive (DR13 MS), all presenting with highly active disease following immunomodulatory treatment with natalizumab, were selected as PBMC donors (*Table 1*). A DR15-positive healthy individual (DR15 HI) was selected as a control donor. Fresh blood samples were used for immunoprofiling by fluorescence-activated cell sorting (FACS) with a panel of standard human immune cell marker antibodies, for screening of plasma antibody responses to viruses, and for the isolation of fresh PBMC for transplantation (*Table 1*; *Figure 1—figure supplement 1*; *Figure 1—figure supplement 1—source data 1C*). Groups of seven to eight B2m-NOG mice were injected intravenously with freshly prepared PBMC ($1 \times 10^7$/mouse) from each donor (*Figure 1—figure supplement 2*, protocol 1). The transplanted mice were monitored for human immune cell engraftment by flow cytometry analysis of small blood samples recovered from the tail vein at different time points and all showed progressive engraftment by human CD45$^+$ (hCD45$^+$) leukocytes from day 7 (*Figure 1A*; *Figure 1—source data 1*; *Figure 1—figure supplement 4A*; *Figure 1—figure supplement 4—source data 1A*). At day 14 post-transplantation (dpt 14), four to five mice from each group were immunized for EAE using a mixture of immunodominant T cell myelin peptide antigens, following two different protocols. In EAE experiment 1, groups of DR13 MS, DR15 MS1, and DR15 HI mice were immunized twice, 7 days apart, using 200 µg each peptide/mouse/injection (*Figure 1—figure supplement 3B, C*; *Figure 1—figure supplement 3—source data 1A*). In EAE experiment 2, groups of DR15 MS2-5 were immunized once, using 100 µg each peptide/mouse (*Figure 1—figure supplement 3C*). Several PBMC donors used in this study were previously shown to have human T cell proliferation responses to myelin peptides (*Dagkonaki et al., 2020*; *Table 1*). Non-immunized mice, and mice immunized for EAE with low-dose peptides ($1 \times 100$ µg) showed steadily increasing levels of blood hCD45$^+$ leukocytes up to sacrifice at dpt 42 (*Figure 1A*; *Figure 1—figure supplement 4A*). Mice immunized for EAE with high-dose peptides ($2 \times 200$ µg) showed reduced levels of blood hCD45$^+$ leukocytes following immunization (*Figure 1A*). Further FACS analysis showed preferential expansion of hCD4$^+$ T lymphocytes compared to hCD8$^+$ T lymphocytes in blood and spleens of all engrafted mice (*Figure 1B, C*; *Figure 1—figure supplement 4B*; *Figure 1—figure supplement 4—source data 1B*), a finding consistent with a previous report describing the B2m-NSG model (*King et al., 2009a*).

Successful engraftment of mice by hCD45$^+$ leukocytes was confirmed in the spleen recovered from non-immunized and immunized mice at sacrifice on dpt 42. Further analysis of splenocytes revealed robust engraftment of hCD4$^+$ and hCD8$^+$ T cells in all mice, and of hCD19$^+$ B cells in DR13 MS, DR15 MS3, and DR15 MS5 mice (*Figure 1C*; *Figure 1—figure supplement 5*; *Figure 1—figure supplement 5—source data 1*). Notably, B cell engraftment was best in MS patients interpreted as having suspected recent or ongoing reactivation of Epstein–Barr virus (EBV), as determined by plasma levels of anti-EBV antibodies (*Table 1*; *Figure 1—figure supplement 1C*; *Figure 1—figure supplement 1—source data 1C*). Human immune cells other than T and B cells, notably monocytes, were undetectable in dpt 42 spleen. Analysis of cytokine production by intracellular staining showed high proportions of interferon-γ (IFN-γ)-producing hCD4$^+$ and hCD8$^+$ splenocytes, and IL-17A-producing hCD4$^+$ and hCD4$^-$ splenocytes in both immunized and non-immunized mice (*Figure 1D*).

To estimate the contribution of mouse myeloid cell populations in the humanized mice, we analyzed peripheral blood from naive B2m-NOG mice, and from naive and CFA-immunized (day 8

**Table 1.** Clinical and demographic data of MS patient and healthy control blood donors.

| HLA genotype Diagnosis (donor type) | G | Age | MS duration (year) | Therapies | Relapse (12/24 months) | MRI Brain/Gd | MRI Spine/Gd | EDSS | Viral infections (titre) | EBV clinical interpretation (Figure 1—figure supplement 1C) | T cell responses [Dagkonaki et al., 2020] |
|---|---|---|---|---|---|---|---|---|---|---|---|
| DR13 1302/1303 RRMS (donor for DR13 MS mice) | F | 42 | 27 | Azathioprine Interferon beta-1a Interferon beta-1b Fingolimod Natalizumab (5 months) Cortisone (for 1 month prior to sampling) | 1/1 Last relapse 05/23 | (+) stable | (+) GD (+) | 2.5 | JC+ (1.24) EBV VCA IgG+ EBV VCA IgM− EBV VCA IgA+ EBV EA IgG+ EBV EBNA1 IgG+ EBV EBNA1 IgM− | Suspect recent or ongoing reactivation | MBP83-99 hMOG35-55 [Patient 17 *Dagkonaki et al., 2020*] |
| DR15 0402/1501 RRMS (donor for DR15 MS1 mice) | F | 38 | 15 | Interferon beta-1a Fingolimod Natalizumab (24 months) To Cladribine (after sampling) | 1/1 No relapse during the last year | (+) stable | (+) stable | 2.5 | JC+ (1.47) EBV VCA IgG+ EBV VCA IgM− EBV VCA IgA− EBV EA IgG− EBV EBNA1 IgG+ EBV EBNA1 IgM+ | Anomalous reactivation | MBP13-32 MBP83-99 hMOG35-55 [Patient 11 *Dagkonaki et al., 2020*] |
| DR15 04/15 healthy (donor for DR15 HI mice) | F | 28 | NA | None | NA | NA | NA | NA | JC ND EBV VCA IgG+ EBV VCA IgM− EBV VCA IgA− EBV EA IgG+ EBV EBNA1 IgG+ EBV EBNA1 IgM+ | Anomalous reactivation | None [Normal 1 *Dagkonaki et al., 2020*] |
| DR15 RRMS (donor for DR15 MS2 mice) | F | 47 | 10 | Dimethyl Fumarate Natalizumab (1 year and ongoing) | 0/0 No relapses during the last year | (+) stable | (+) stable | 1.5 | JC+ (0,96) EBV VCA IgG+ EBV VCA IgM− EBV VCA IgA− EBV EA IgG− EBV EBNA1 IgG+ EBV EBNA1 IgM− | Seropositive without symptoms of active infection | ND |
| DR15 RRMS (donor for DR15 MS3 mice) | F | 39 | 15 | Interferon beta-1b Natalizumab (8 years and ongoing) | 0/0 No relapse during the last year | (+) stable | (+) stable | 2.0 | JC (−) EBV VCA IgG+ EBV VCA IgM− EBV VCA IgA+ EBV EA IgG+ EBV EBNA1 IgG+ EBV EBNA1 IgM− | Suspect recent or ongoing reactivation | ND |
| DR15 RRMS (donor for DR15 MS4 mice) | F | 53 | 23 | Interferon beta-1a Interferon beta-1b Glatiramer Acetate Natalizumab (13 years and ongoing) | 0/0 No relapse during the last year | (+) stable | (+) stable | 3.5 | JC (−) EBV VCA IgG+ EBV VCA IgM− EBV VCA IgA− EBV EA IgG+ EBV EBNA1 IgG+ EBV EBNA1 IgM+ | Anomalous reactivation | ND |
| DR15 RRMS (donor for DR15 MS5 mice) | F | 30 | 13 | Interferon beta-1a Interferon beta-1b Fingolimod Natalizumab (8 years and ongoing) | 0/0 No relapse during the last year | (+) stable | (+) stable | 1.5 | JC (−) EBV VCA IgG+ EBV VCA IgM+ EBV VCA IgA+ EBV EA IgG− EBV EBNA1 IgG+ EBV EBNA1 IgM− | Suspect recent or ongoing reactivation | ND |

KEY: EBV, Epstein-Barr virus; EDSS, expanded disability status scale; G, gender; Gd, gadolinium-enhancing lesions; JC; John Cunningham virus; NA, not applicable; ND, not done; RRMS, relapse-remitting MS.

post-immunization) NOD-*scid* parental strain and wild-type C57BL/6 (B6) mice, by flow cytometry. Immature Ly6C$^{hi}$ monocytes and Ly6G$^+$ cells are known to massively expand in the periphery of B6 mice following immunization with the adjuvants used for EAE, and are critical for the development of EAE (*King et al., 2009b*; *Dagkonaki et al., 2022*). Unlike B6 mice, very high proportions of mouse (m) CD11b$^+$, mCD11b$^+$Ly6C$^{hi}$, and mCD11b$^+$Ly6G$^+$ myeloid cells were already present in the

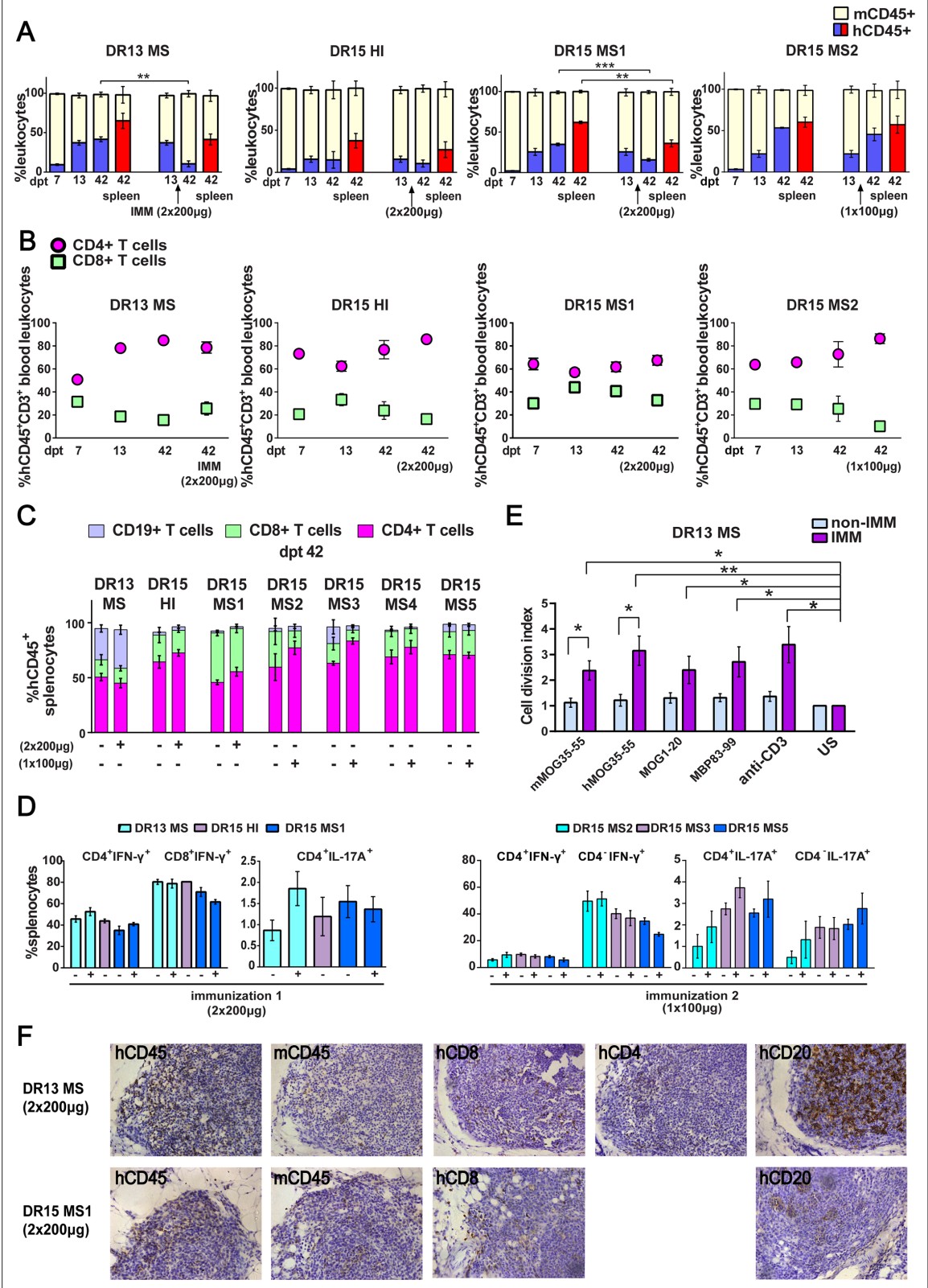

**Figure 1.** Reconstitution of a human adaptive immune system in B2m-NOG mice engrafted with peripheral blood mononuclear cell (PBMC) from multiple sclerosis (MS) patient and healthy donors. (**A**) Progressive engraftment of human (h) CD45+ leukocytes from selected donors in groups of B2m-NOG mice, non-immunized (left bars) and immunized for experimental autoimmune encephalomyelitis (EAE) (right bars) using a myelin peptide cocktail in repeat immunizations with 200 μg/each myelin peptide (EAE experiment 1), or single immunization with 100 μg/each myelin peptide (EAE experiment

*Figure 1 continued on next page*

*Figure 1 continued*

2), measured in peripheral blood samples taken at different time points , and in spleen recovered at sacrifice 42 days' post-transplantation (dpt 42) by fluorescence-activated cell sorting (FACS) (non-immunized mice, dpt 7 & 13, n=7-8 mice/group; dpt 42, n=3 mice/group; immunized mice, dpt 42, n=4-5 mice/group). (**B**) Proportions of hCD4$^+$ and hCD8$^+$ T cells in blood hCD45$^+$CD3$^+$ T cells at different time points by FACS (non-immunized mice, dpt 7 & 13, n=7-8 mice/group; dpt 42, n=3 mice/group; immunized mice, dpt 42, n=4-5 mice/group). (**C**) Proportions of human immune cell subpopulations in spleens of immunized and non-immunized mice at dpt 42 by FACS (non-immunized mice n=3 mice/group; immunized mice n=4-5 mice/group). (**D**) Proportions of interferon-γ- and IL-17A-producing CD4$^+$ and CD8$^+$ (or CD4$^-$) T cells in splenocytes recovered from mice in the different groups at dpt 42 by FACS (non-immunized mice n=2-3 mice/group; immunized mice n=4-5 mice/group). (**E**) Antigen-specific T cell proliferation responses to the immunizing antigens (mMOG35-55, hMOG35-55, MOG1-20, and MBP83-99), anti-hCD3 (positive control) and medium (unstimulated; US), in splenocytes recovered from non-immunized and immunized DR13 MS PBMC humanized mice at dpt 42 by FACS (non-immunized mice n=3 mice/group; immunized mice n=3-4 mice/group). Results are expressed as a cell division index. (**F**) Draining lymphoid structures in inguinal fat of mice engrafted with DR13 MS and DR15 MS PBMC and immunized with myelin peptides, recovered at dpt 42. Immunohistochemistry revealed the presence of hCD45- and mCD45-positive leukocytes, hCD4- and hCD8-positive T cells, and hCD20-positive B cells (latter only DR13 MS mice). Scale bar 200 µm; ×20 objective (**F**). All results are depicted as mean ± standard error of the mean (SEM). Statistical significance is shown after pairwise comparisons between groups using Student's *t* test (*p ≤ 0.05, **p ≤ 0.01, ***p ≤ 0.001).

The online version of this article includes the following source data and figure supplement(s) for figure 1:

**Source data 1.** Proportions of human and mouse immune cell subpopulations in peripheral blood and spleen of humanized B2m-NOG mice at different time points measured by FACS.

**Figure supplement 1.** Analyses of peripheral blood samples from multiple sclerosis (MS) and healthy peripheral blood mononuclear cell (PBMC) donors.

**Figure supplement 1—source data 1.** C: Levels of EBV infection markers measured in the plasma of MS and healthy individual blood donors by ELISA.

**Figure supplement 2.** Optimization of human peripheral blood mononuclear cell (PBMC) isolation protocol.

**Figure supplement 3.** Experimental autoimmune encephalomyelitis (EAE) protocols used in C57BL/6 and humanized B2m-NOG mice.

**Figure supplement 3—source data 1.** A: Clinical scores of individual C57BL/6 mice immunized for EAE with the different peptide protocols shown in B.

**Figure supplement 4.** Reconstitution of a human adaptive immune system in B2m-NOG mice engrafted with peripheral blood mononuclear cell (PBMC) from multiple sclerosis (MS) patiients.

**Figure supplement 4—source data 1.** Proportions of CFSE-low (highly proliferating) splenocytes isolated from the different B2m-NOG humanized mice at the end of the experiment (dpt 42), stimulated with myelin peptides or polyclonal stimuli ex vivo, and measured by FACS.

**Figure supplement 5.** Human immune cell engraftment in peripheral blood mononuclear cell (PBMC) humanized B2m-NOG mouse splenocytes.

**Figure supplement 5—source data 1.** Proportions of human immune cell subpopulations in splenocytes isolated from humanized B2m-NOG mice at the end of the experiment (dpt 42) and measured by FACS.

**Figure supplement 6.** Comparative fluorescence-activated cell sorting (FACS) analysis of mouse CD11b$^+$ myeloid cell subpopulations in the peripheral blood of non-peripheral blood mononuclear cell (PBMC)-engrafted mouse strains.

**Figure supplement 6—source data 1.** Proportions of mouse myeloid cell subpopulations in peripheral blood of non-PBMC engrafted B2m-NOG, NOD-scid and C57BL/6 mice, either non-immunized (naive) or immunized with CFA and measured by FACS.

blood of naive B2m-NOG and NOD-*scid* mice, and were not significantly increased by immunization (*Figure 1—figure supplement 6*; *Figure 1—figure supplement 6—source data 1*).

Functionality of the human immune system in B2m-NOG mice was investigated using an ex vivo proliferation assay for T cell responses in splenocytes (*Dagkonaki et al., 2022*), and a cell-based assay for anti-MOG antibody responses in plasma. Briefly, splenocytes recovered from non-immunized and immunized mice at sacrifice were stimulated ex vivo individually with the four myelin peptides used for immunization (mMOG35-55, hMOG35-55, MOG1-20, and MBP83-99). T cell proliferation responses were measured using a CFSE dilution assay. Splenocytes isolated from DR13 MS mice (*Figure 1E*), showed significant T cell proliferation responses to all myelin peptides, equal to control splenocytes stimulated with anti-hCD3 antibody. Notably, splenocytes from DR15 MS and DR15 HI mice showed limited or absence of T cell proliferation responses to myelin peptides or polyclonal T cell stimuli, anti-hCD3 antibody and phytohaemagglutinin (PHA) (*Figure 1—figure supplement 4C*; *Figure 1—figure supplement 4—source data 1C*). The difference in responses between DR13 and DR15 donors cannot be explained by differences in human B cell engraftment, which are potential antigen-presenting cells in the ex vivo T cell proliferation assay, because B cell engraftment was seen in both DR13 MS and DR15 MS mice (*Figure 1C*). A previous study showed that T cells isolated from DR15 MS patients display high levels of autoproliferation (*Jelcic et al., 2018*). Therefore, it is possible

that poor human T cell proliferation responses in DR15 mice might result from inherently high T cell autoproliferation in the DR15 MS donor PBMC prior to engraftment, although this possibility needs to be formally tested. Anti-hMOG IgG antibody responses were not detectable in the plasma of any mice at any time point.

We next investigated whether draining lymph node (LN) structures could be identified in the immunized mice at sacrifice. Mice lacking the common cytokine receptor γ-chain gene are known to have poor lymphoid organ structure and functioning, because NK cells are needed for the formation of LN inducer cells, which in turn are needed for LN organogenesis (*Mebius, 2003*). Nevertheless, the inguinal fat tissue from several immunized mice showed accumulated masses of leukocytes containing mCD45- and hCD45-positive leukocytes, CD8-, CD4-, and CD20-positive lymphocytes by immuno-histochemistry, although lacking obvious LN organization (*Figure 1F*). Together the results show that human T and B lymphocytes are efficiently engrafted in B2m-NOG mice by PBMC from MS patients, that functional T cell responses can be detected depending upon the PBMC donor, while lack of organized LN and germinal centers prevents the generation of anti-MOG IgG antibody responses.

## Human T cells accumulate at CNS borders in non-immunized DR15 MS and DR15 HI mice, and form spontaneous parenchymal lesions in brain and spinal cord of DR15 MS mice

For the period of the study, none of the engrafted mice showed clinical symptoms of neuroinflammation using criteria commonly used for scoring EAE (*Dagkonaki et al., 2020*), or other neurological deficits (*Guyenet et al., 2010*). Mild symptoms of GVHD, specifically fur ruffling and reduced mobility (*Cooke et al., 1996*), were recorded in few mice independent of group close to the time of sacrifice. To determine whether human immune cells enter CNS tissues in PBMC humanized B2m-NOG mice, we performed immunohistochemical analyses of serial sections from brain (*Figure 2*; *Figure 2— source data 1*) and spinal cord (*Figure 3*; *Figure 3—source data 1*) recovered from non-immunized mice at sacrifice using marker antibodies for human and mouse immune cells as well as for mouse microglia and astrocytes. To further monitor GVHD development we also performed standard histological analyses of lung and liver, two main tissue targets of GVHD.

DR13 MS humanized mice showed very few hCD45-positive leukocytes at CNS borders, specifically meninges, and none in CNS parenchyma, and no further analysis was made (data not shown). DR15 HI mice showed an accumulation of human immune cells at CNS borders, particularly the meninges and some in the choroid plexus of the interventricular foramen between lateral and third ventricles of the brain, and few cells scattered throughout the brain parenchyma (*Figure 2A–C*). Both brain border and parenchymal hCD45-positive immune cells comprised mainly hCD8-positive T cells (*Figure 2D*), leading to low hCD4/hCD8 T cell ratios, especially in parenchyma (*Figure 2E*). Mild activation of Iba1-positive microglia and GFAP-positive astrocytes was seen at sites of immune cell accumulation at CNS borders (*Figure 2C*). In spinal cord, hCD3-positive T cells accumulated at borders and few individual cells infiltrated parenchyma of white and gray matter (*Figure 3A–C*), forming rare small T cell lesions (≥3 adjacent cells) in the white matter (*Figure 3A*, bottom left panel, arrowhead). Spinal cord-infiltrating cells in DR15 HI mice also comprised mainly hCD8-positive T cells, leading to low hCD4/hCD8 T cell ratios (*Figure 3D*). Mouse CD45-positive cells were rare at barriers, absent from parenchyma, and demyelination was not observed in the brain or spinal cord of DR15 HI mice.

Three of the five groups of DR15 MS mice, specifically DR15 MS1, DR15 MS3, and DR15 MS5 mice, showed severe spontaneous CNS pathology with accumulation of human immune cells at borders, particularly the meninges and choroid plexus of the interventricular foramen between lateral and third ventricles, widespread immune cell infiltration of the parenchyma, and T cell lesions in the brain, brainstem, and spinal cord. Brain border and parenchymal hCD45-positive immune cells comprised mainly hCD8-positive T cells (*Figure 2A–D*), leading to low hCD4/hCD8 T cell ratios, especially in the parenchyma (*Figure 2E*). These mice showed numerous spontaneous parenchymal hCD45-positive cell lesions (≥3 adjacent cells). The location and severity of parenchymal lesions were different in each patient, and involved brainstem (DR15 MS1; *Figure 2A, F*), hippocampus (DR15 MS3; *Figure 2A, Gi, ii*), optic chiasm (DR15 MS3; *Figure 2A, H*), and sub-hippocampal regions (DR15 MS3; *Figure 2A, Giii*, DR15 MS5; *Figure 2A, I, J*). Locally activated Iba1-positive microglia and GFAP-positive astrocytes were observed at sites of immune cell accumulation (*Figure 2C*). DR15 MS2 and DR15 MS4 mice showed only few human immune cells at borders and rare parenchymal lesions. In the spinal cord,

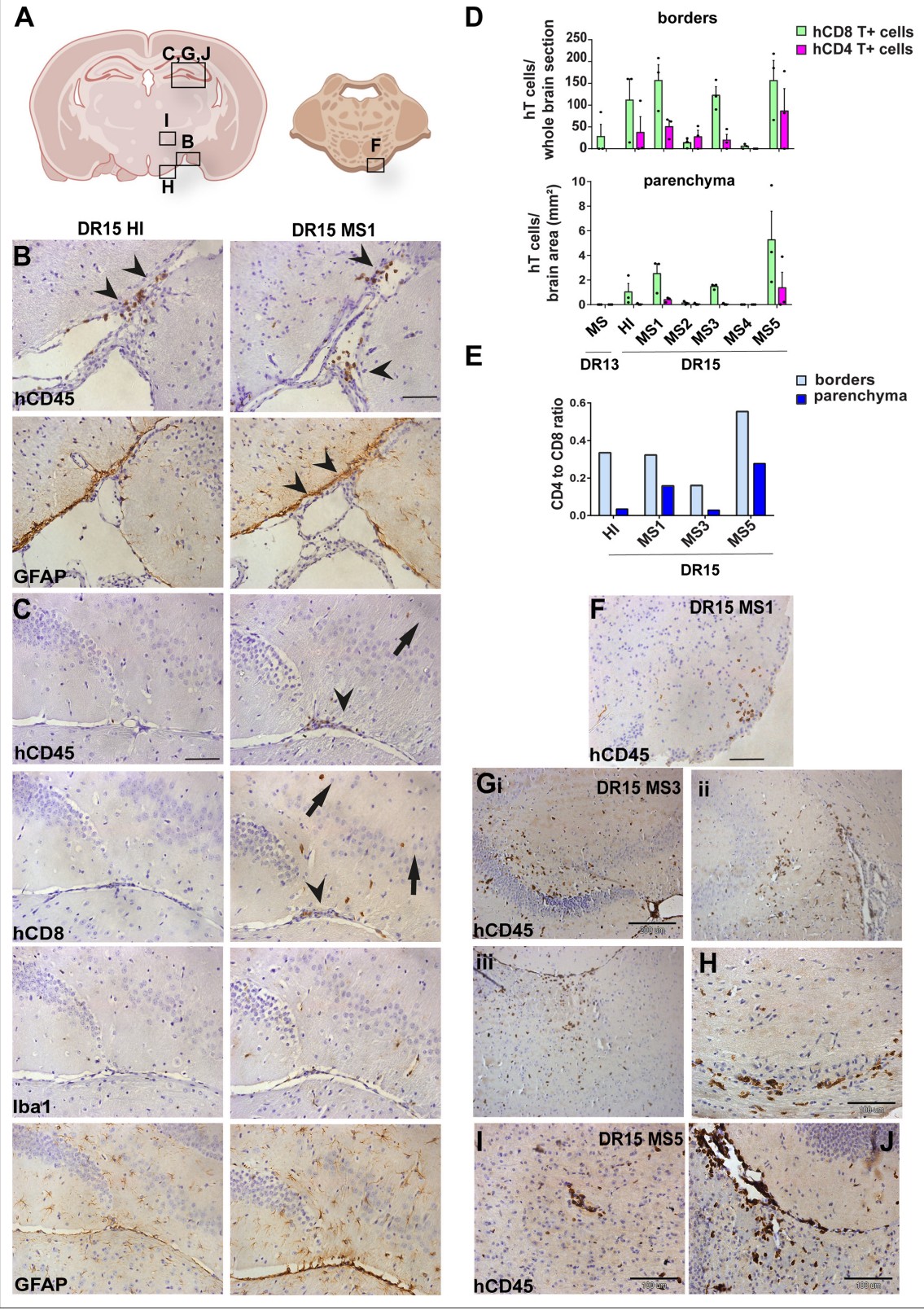

**Figure 2.** Human T cells accumulate at brain borders in non-immunized DR15 MS and DR15 HI mice, and form spontaneous parenchymal lesions in brain of DR15 MS mice. Immunohistochemical analysis of the brain from non-immunized peripheral blood mononuclear cell (PBMC) B2m-NOG mice showing infiltration by human (h) and mouse (m) CD45-positive leukocytes, and hCD8-positive T cells, in brain border and parenchymal regions (denoted in diagrams of brain and brainstem **A**). (**B, C**) Accumulation of hCD45- and mCD45-positive leukocytes and hCD8-positive T cells, together with local

*Figure 2 continued on next page*

*Figure 2 continued*

activation of Iba1-positive microglia and glial fibrillary acid protein (GFAP)-positive astrocytes, at border regions in the brains of DR15 HI (left panels) and DR15 MS (right panels) mice, specifically at meninges close to the optic tract (B, arrowheads) and in the connective tissue of the interventricular foramen joining the lateral and third ventricles (C, arrowheads). Scattered hCD45-positive leukocytes and hCD8 T cells in brain parenchyma of DR15 MS mice (C, arrows). (**D**) Counting of border-associated and parenchymal hCD8- and hCD4-positive T cells in whole coronal sections of brain of humanized mice, represented as total cells and cells/mm² (n=3 mice/group) (**E**) Ratios of hCD4/hCD8 T cells at borders and parenchyma of selected humanized mice. (**F**) Small hCD45-positive immune cell lesion in brainstem of DR15 MS1 mice (see also diagram A). (**G**) hCD45-positive immune cell lesions in gray matter of hippocampus (**i, ii**) and sub-hippocampus/thalamus (**iii**), and white matter of optic chiasm (**H**) of DR15 MS3 mice. (**I, J**) hCD45-positive immune cell lesions in gray matter of thalamus and hippocampus, respectively, of DR15 MS5 mice. Scale bars 100 μm; ×20 objective (**B, C, F**), 200 μm; ×10 objective (**G**), 100 μm; ×40 objective (**H–J**). All results are depicted as mean ± standard error of the mean (SEM). Statistical analysis was performed by pairwise comparisons between different groups of mice using Student's *t* test.

The online version of this article includes the following source data and figure supplement(s) for figure 2:

**Source data 1.** Numbers of human CD4 and CD8 T cells counted at borders (total cells in comparable whole coronal brain sections) and in parenchyma (cells/mm2) (D), as well as the human CD4/CD8 T cell ratios derived from these measurements (E).

**Figure supplement 1.** Inflammation in peripheral graft versus host disease (GVHD) target tissues in humanized B2m-NOG mice.

**Figure supplement 1—source data 1.** Inflammation scores in the liver and lung of non-immunized and immunized humanized B2m-NOG mice, as measured by semi-quantitative analysis of H&E stained sections isolated at the end of the experiment.

hCD3-positive T cells accumulated at borders and infiltrated both white and gray matter parenchyma, forming lesions in both areas (*Figure 3A*, right panels, arrowheads). Semi-quantitative analysis of parenchymal hCD3-positive T cells in comparable whole spinal cord sections showed a tendency for increased cells in DR15 MS1 mice compared to control HI mice (*Figure 3B*), and numbers of T cell lesions in the gray matter were significantly increased (*Figure 3Ci*). Unlike control HI mice, DR15 MS1 mice showed numerous small T cell lesions in the gray matter, often close to the central canal suggesting this is a main site of human T cell entry into the spinal cord gray matter (*Figure 3A*, right top panel, arrowheads). Spinal cord-infiltrating immune cells in DR15 MS1 mice also comprised mainly hCD8-positive T cells, leading to low hCD4/hCD8 T cell ratios (*Figure 3D*). In contrast to hCD45-positive immune cells that were scattered in spinal cord parenchyma (*Figure 3E*), mouse CD45-positive cells were rare at barriers, absent from parenchyma, and demyelination was not observed in the brain or spinal cord of any DR15 MS mice.

To determine whether differences in the severity of CNS immune infiltration in the immunized mice could be secondary to levels of GVHD inflammation in the periphery, we performed semi-quantitative analysis of hematoxylin and eosin (H&E) stained liver and lung sections, which are main targets of GVHD responses (*Cooke et al., 1996*). The mouse groups showed equal levels of inflammation in both liver and lung, revealing a dissociation between peripheral GVHD and CNS inflammation (*Figure 2—figure supplement 1*; *Figure 2—figure supplement 1—source data 1*). These results show that non-immunized DR15 MS1 humanized mice show increased infiltration of brain and spinal cord by human T cells compared to non-immunized DR15 HI mice, and uniquely develop spontaneous T cell lesions in spinal cord gray matter and brain parenchyma.

## Immunization with myelin peptides increases hCD4 T cell infiltration of CNS parenchyma resulting in mixed hCD4/hCD8 T cell lesions in brain and spinal cord of DR15 MS and DR15 HI mice

To further investigate whether human immune cells have the potential to induce CNS immunopathology in PBMC humanized B2m-NOG mice, we immunized mice for EAE. In a previous study, humanized HLA-DR2b transgenic mice lacking all mouse MHCII genes were found to require higher amounts of peptide antigen than B6 mice to induce clinical EAE (*Dagkonaki et al., 2020*). Here, EAE tests in wild-type B6 mice showed that disease could be induced by immunization with high amounts of peptides with no evidence of increased morbidity or mortality (*Figure 1—figure supplement 3A, B*). A peptide cocktail containing myelin epitopes previously associated with MS, specifically hMOG35-55, MOG1–20 and MBP83-99 (*Pette et al., 1990*; *Ota et al., 1990*; *Kerlero de Rosbo et al., 1993*; *Cao et al., 2015*), plus mMOG35-55, was chosen for immunization in groups of 12- to 14-week-old humanized B2m-NOG mice on dpt 14. In the first experiment, groups of B2m-NOG mice engrafted with PBMC from DR13 MS, DR15 MS1 and DR15 HI donors received a repeat immunization with 200 μg/peptide spaced 7 days apart, similar to that used in HLA-DR2b transgenic mice. In

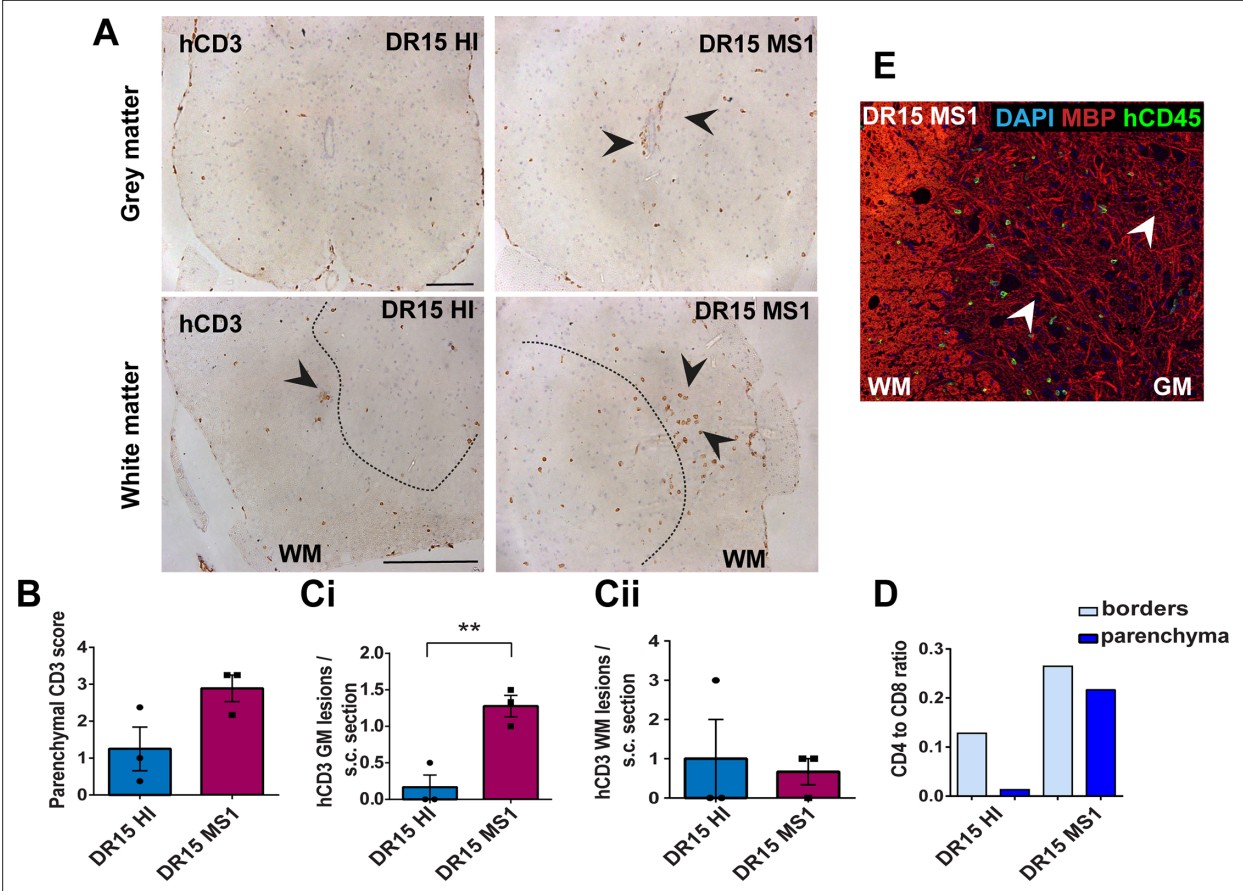

**Figure 3.** Human T cells infiltrate spinal cord white matter in non-immunized DR15 MS and DR15 HI mice, and form gray matter lesions in DR15 MS mice. Immunohistochemical analysis of spinal cord from non-immunized peripheral blood mononuclear cell (PBMC) B2m-NOG mice showing infiltration by human (h) CD3-positive T cells in the gray and white matter regions. (**A**) Infiltrating hCD3-positive T cells were scattered individually throughout spinal cord gray and white matter and formed small lesions in the white matter (WM) of both DR15 MS and DR15 HI mice (lower panels, arrowheads), and gray matter lesions only in DR15 MS mice (upper panels, arrowheads). The dotted lines mark the boundary between gray matter and white matter. (**B**) Semi-quantitative estimation of hCD3-positive T cells in comparable whole spinal cord sections from DR15 HI and DR15 MS mice (n=3 mice/group). (**C**) Counting of parenchymal hCD3-positive T cell lesions (≥3 adjacent cells) in gray matter (GM) (i) and white matter (WM) (ii) in whole spinal cord sections from DR15 HI and DR15 MS mice. (**D**) Ratios of hCD4/hCD8 T cells at borders and parenchyma of DR15 HI and DR15 MS mice. (**E**) Double immunofluorescence staining for hCD45-positive leukocytes (green, arrowheads) and MBP-positive myelin (red), with 4',6-diamidino-2-phenylindole (DAPI) counterstained nuclei (blue), in spinal cord WM of DR15 MS mice. Scale bars 100 µm; ×20 objective (A, top panels); ×40 objective (A, bottom panels). All results are depicted as mean ± standard error of the mean (SEM). Statistical analysis was performed by pairwise comparisons between different groups of mice using Student's $t$ test (**$p \leq 0.01$).

The online version of this article includes the following source data for figure 3:

**Source data 1.** Semi-quantitative analysis of human CD3-positive T cells (B) and numbers of human CD3-positive lesions (defined as 3 or more adjacent cells) in grey matter (GM) and white matter (WM) regions (C) in comparable whole spinal cord sections from DR15 HI and DR15 MS mice, as well as human CD4/CD8 T cell ratios at borders and in parenchyma of these sections.

the second experiment, groups of B2m-NOG mice engrafted with PBMC from DR15 MS2-5 donors received a single immunization with 100 µg/peptide, to reduce the possibility of T cell tolerance (*Figure 1—figure supplement 3C*). None of the PBMC humanized B2m-NOG mice immunized with myelin antigens developed typical clinical symptoms of EAE, a finding consistent with a previous study in NSG mice humanized with PBMC from healthy donors (*Zayoud et al., 2013*).

To investigate whether immunization for EAE increased immune cell infiltration of the brain (*Figure 4*; *Figure 4—source data 1*) and spinal cord (*Figure 5*; *Figure 5—source data 1*) we performed immunohistochemical analysis of tissues recovered from the mice at sacrifice at dpt 42. Immunized DR13 MS mice and their non-immunized controls showed very few hCD45-positive leukocytes at CNS borders and infiltrating spinal cord parenchyma, so no further analysis was made (data not shown). Immunized

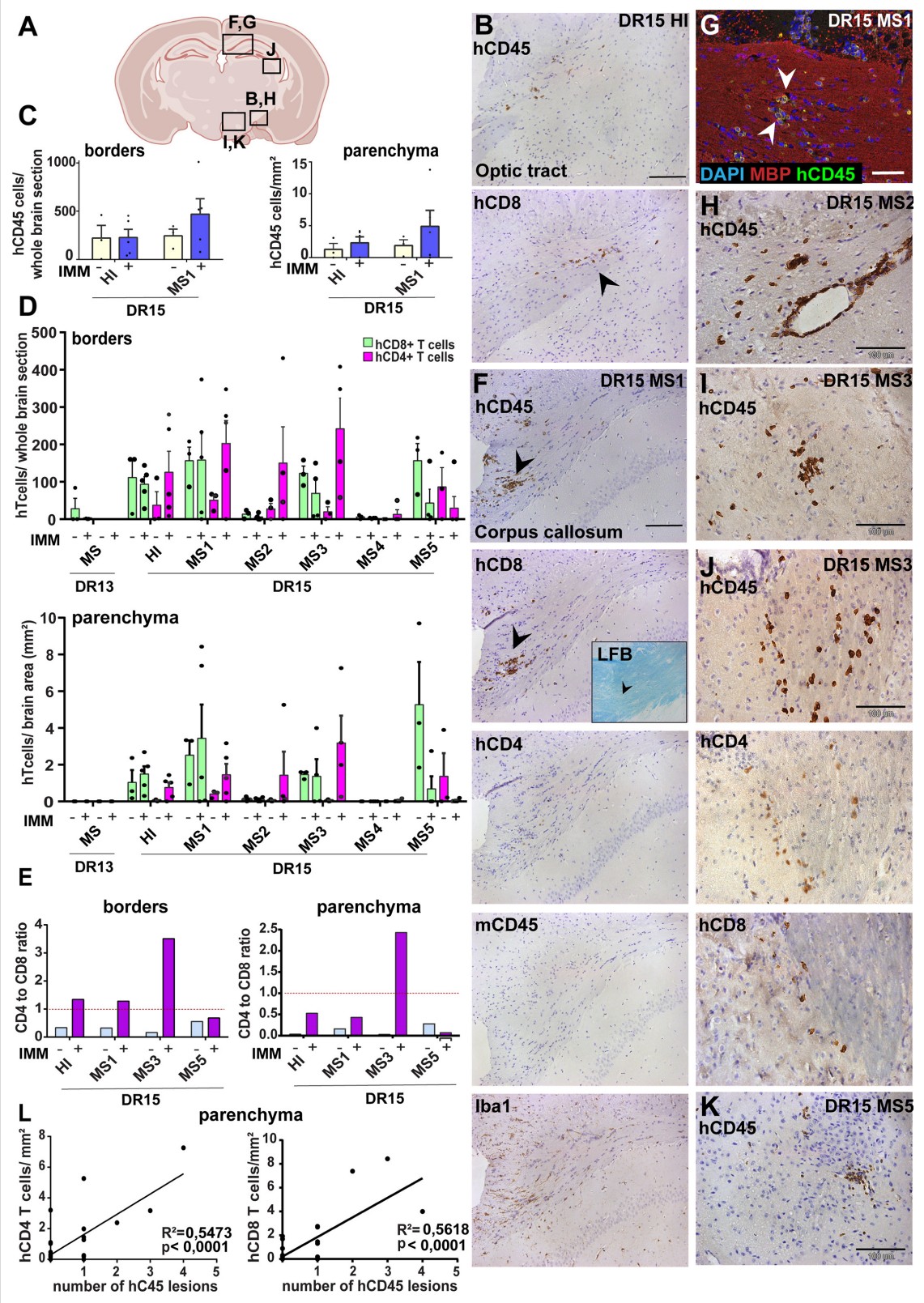

**Figure 4.** Immunization with myelin peptides increases hCD4 T cell infiltration of brain parenchyma resulting in mixed hCD4/hCD8 T cell lesions in brain of DR15 MS and DR15 HI mice. Immunohistochemical analysis of the brain from peripheral blood mononuclear cell (PBMC) B2m-NOG mice immunized for experimental autoimmune encephalomyelitis (EAE) showing infiltration by human (h) and mouse (m) CD45-positive leukocytes, hCD8- and hCD4-positive T cells, and local activation of Iba1-positive microglia and GFAP-positive astrocytes, in brain regions denoted in the diagram (**A**). (**B**) Individual

*Figure 4 continued on next page*

*Figure 4 continued*

hCD45-positive leukocytes and hCD8-positive T cells form small lesions in the optic tract (arrowhead) in immunized DR15 HI mice. (**C**) Counting of hCD45-positive immune cells at borders (total cells in section) and parenchyma (cells/mm²) in whole coronal sections of brain from non-immunized (n=3/group) and immunized (n=5 mice/group) DR15 HI and DR15 MS mice. (**D**) Counting of hCD4- and hCD8-positive T cells at borders (total cells in section) and parenchyma (cells/mm²) in whole coronal sections of brain from non-immunized (n=3 mice/group) and immunized DR13 MS, DR15 HI and DR15 MS1 (all n=5 mice/group), DR15 MS2-5 (n=4 mice/group). (**E**) Ratios of hCD4/hCD8 T cells at borders and parenchyma of non-immunized and immunized DR15 HI and DR15 MS mice. (**F**) Prominent lesions in the corpus callosum white matter of two of five DR15 MS1 mice (arrowheads), containing hCD45- and mCD45-positive leukocytes, hCD4- and hCD8-positive T cells, and locally activated Iba1-positive microglia. Inset shows a serial section stained by Luxol fast blue showing absence of demyelination. (**G**) Double immunofluorescence staining for hCD45-positive leukocytes (green, arrowheads) and MBP-positive myelin (red), with DAPI counterstained nuclei (blue), in corpus callosum in immunized DR15 MS1 mice, showing inflammatory lesion without demyelination. (**H**) Small white matter lesion in DR15 MS2 mouse. (**I**) Prominent white and (**J**) gray matter lesions containing both hCD4 and hCD8 T cells in DR15 MS3 mice. (**K**) Small lesion containing human hCD45-positive immune cells in sub-thalamic area of DR15 MS5 mice. (**L**) Correlation analysis between numbers of hCD4 or hCD8 with number of hCD45 lesions in brain parenchyma of all combined immunized DR15 HI and DR15 MS1-5 mice. Scale bars 100 μm; ×20 objective (**B, F**); ×40 objective (**G, K**). All results are depicted as mean ± standard error of the mean (SEM). Statistical analysis was performed by pairwise comparisons between different groups of mice using Student's *t* test.

The online version of this article includes the following source data for figure 4:

**Source data 1.** Numbers of human CD45-positive immune cells counted at brain borders (total cells in comparable whole coronal brain sections) and in parenchyma (cells/mm2) (C), as well as of human CD8 and human CD4 T cell-positive cells counted at brain borders (total cells in comparable whole coronal brain sections) and in parenchyma (cells/mm2) (D) of non-immunized and immunized humanized B2m-NOG mice, as well as the human CD4/CD8 T cell ratios derived from these measurements (E).

DR15 HI mice showed accumulation of hCD45-positive leukocytes at the borders, particularly the meninges and choroid plexus, and few cells scattered throughout the parenchyma (*Figure 4A–C*). Brain border and parenchymal human immune cells comprised hCD8-positive T cells, and in contrast to non-immunized controls also contained high numbers of hCD4-positive cells (*Figure 4D*), leading to high hCD4/hCD8 T cell ratios (*Figure 4E*). This finding is consistent with successful activation of hCD4-positive T cells by the EAE immunization protocol. In one of five immunized DR15 HI mice, a small T cell lesion was detected in the optic tract (*Figure 4A, B*, arrowhead). Locally activated Iba1-positive microglia and GFAP-positive astrocytes were present at sites of immune cell accumulation at borders and in optic tract lesions (data not shown). In spinal cord, hCD3-positive T cells accumulated at borders and individual cells infiltrated parenchyma of gray and white matter (*Figure 5A*, left panels, *Figure 5B*), forming small T cell lesions in the white matter (*Figure 5A*, bottom left panel, arrowhead, *Figure 5Cii*). In the spinal cord of immunized DR15 HI mice, border and parenchymal T cells comprised hCD8-positive T cells and also higher numbers of hCD4-positive T cells compared to non-immunized controls, leading to higher hCD4/hCD8 T cell ratios (*Figure 5D*). Mouse CD45-positive cells were rare at barriers, absent from parenchyma, and demyelination was not observed in the brain or spinal cord of immunized DR15 HI mice.

Immunized DR15 MS1-5 mice generally showed prominent accumulation of human immune cells at CNS borders, widespread immune cell infiltration of the parenchyma and development of T cell lesions in the brain and the spinal cord. Brain border and parenchymal hCD45-positive leukocytes comprised hCD8-positive T cells together with higher numbers of hCD4-positive cells compared to non-immunized controls (*Figure 4C and D*), leading to higher hCD4/hCD8 T cell ratios (*Figure 4E*). Border and parenchymal T cells were generally increased in immunized DR15 MS mice compared to immunized DR15 HI mice, but differences did not reach significant due to intra-group variation (*Figure 4D*). Unique features of immunized DR15 MS mice included large human immune cell lesions in the corpus callosum white matter of DR15 MS1 mice containing both hCD8- and hCD4-positive T cells (*Figure 4A, F*), without detectable myelin damage (*Figure 4F*, inset, *Figure 4G*), a white matter lesion in DR15 MS2 mice (*Figure 4A, H*), numerous white and gray matter lesions in DR15 MS3 mice (*Figure 4A, I, J*), and small subthalamic gray matter lesions in DR15 MS5 mice (*Figure 4A, K*). Increased brain infiltration by CD4-positive T cells in immunized mice was not associated with increased frequency of brain lesions, but rather the presence of mixed lesions containing both CD4- and CD8-positve T cells (*Figure 4J*). Correlation analysis showed a positive correlation between numbers of hCD4 and hCD8 with numbers of hCD45 lesions (*Figure 4L*). Locally activated Iba1-positive microglia and GFAP-positive astrocytes were present at sites of human immune cell entry at CNS borders and in T cell lesions (*Figure 4F*, and data not shown). In spinal cord, hCD3-positive T cells accumulated at borders and individual cells infiltrated parenchyma of gray and white matter (*Figure 5A*, right panels,

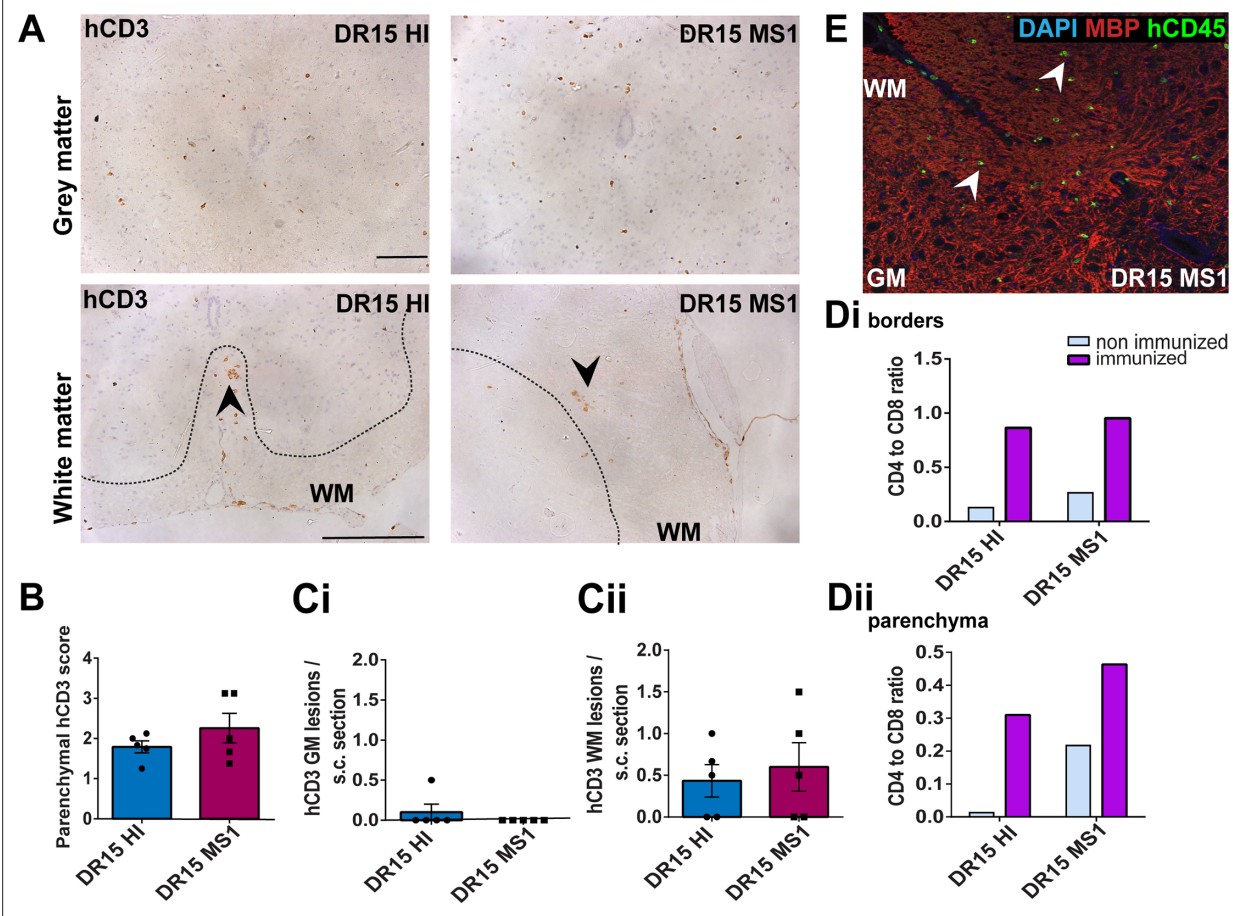

**Figure 5.** Immunization with myelin peptides increases hCD4 T cell infiltration of spinal cord white matter in both DR15 MS and DR15 HI mice. Immunohistochemical analysis of spinal cord from peripheral blood mononuclear cell (PBMC) B2m-NOG mice immunized for experimental autoimmune encephalomyelitis (EAE) showing infiltration by human (h) CD3-positive T cells in the gray and white matter regions. (**A**) Infiltrating hCD3-positive T cells scattered throughout spinal cord gray and white matter and forming small lesions in the white matter (WM) of both DR15 MS1 and DR15 HI mice (bottom panels, arrowheads) The dotted lines mark the boundary between gray and white matter. (**B**) Semi-quantitative estimation of hCD3-positive T cells in comparable whole spinal cord sections from DR15 HI and DR15 MS mice (n=5 mice/group). (**C**) Counting of hCD3-positive T cell lesions (≥3 adjacent cells) in gray matter (GM) (**i**) and white matter (WM) (**ii**) in whole spinal cord sections from DR15 HI and DR15 MS1 mice (n=5 mice/group). (**D**) Ratios of hCD4/hCD8 T cells at borders (**i**) and in parenchyma (**ii**) of spinal cord in non-immunized and immunized DR15 HI and DR15 MS mice. (**E**) Double immunofluorescence staining for hCD45-positive leukocytes (green, arrowheads) and MBP-positive myelin (red), with DAPI counterstained nuclei (blue), in white matter (WM) of immunized DR15 MS spinal cord. Scale bars 100 μm; ×40 objective (**A**). All results are depicted as mean ± standard error of the mean (SEM). Statistical analysis was performed by pairwise comparisons between different groups of mice using Student's *t* test.

The online version of this article includes the following source data for figure 5:

**Source data 1.** Semi-quantitative analysis of human CD3-positive T cells (B), and numbers of human CD3-positive lesions (defined as 3 or more adjacent cells) in grey matter (GM) and white matter (WM) regions (C), in comparable whole spinal cord sections from DR15 HI and DR15 MS mice, as well as human CD4/CD8 T cell ratios at borders and in parenchyma of these sections in non-immunized and immunized humanized B2m-NOG mice.

*Figure 5B*), forming small T cell lesions only in the white matter (WM) (*Figure 5A*, right bottom panel, arrowhead, *Figure 5Cii*). Semi-quantitative analysis of parenchymal hCD3-positive T cells in comparable whole spinal cord sections showed equal infiltration in DR15 MS1 mice compared to control HI mice (*Figure 5B*) and numbers of T cell lesions in the white matter were equal to those in immunized HLA-DR15 HI mice (*Figure 5Cii*). Spinal cord border and parenchymal T cells of immunized DR15 MS1 mice comprised hCD8-positive T cells and also high numbers of hCD4-positive T cells compared to non-immunized controls, leading to higher hCD4/hCD8 T cell ratios (*Figure 5D*). Mouse CD45-positive cells were rare at barriers, absent from parenchyma, and demyelination was not observed in the brain (*Figure 4F*) or spinal cord of immunized DR15 MS mice.

To determine whether CNS immune infiltration in the immunized mice could be secondary to levels of GVHD inflammation in the periphery, we performed semi-quantitative analysis of liver and lung sections with H&E staining. Immunized DR13 MS mice showed the lowest levels of immune cell infiltration in liver and lung, DR15 MS1 mice intermediate levels, and DR15 HI highest levels (*Figure 2—figure supplement 1*; *Figure 2—figure supplement 1—source data 1*). Results revealed an inverse relationship between peripheral and CNS inflammation in the DR15 HI and MS mice, with DR15 HI mice showing highest peripheral and relatively lower CNS involvement, while DR15 MS mice showed the highest CNS and relatively lower peripheral inflammation. These results show that immunization of DR15 HI and DR15 MS mice with myelin peptides equally induced spinal cord white matter inflammatory lesions, and uniquely induced large confluent brain white and gray matter lesions in DR15 MS mice. Taken together, our results show that PBMC from DR15-positive MS patients show increased propensity to induce inflammatory CNS lesions compared to PBMC from a DR13-positive MS patient or a DR15-positive healthy individual.

## Discussion

Here, we show that PBMC from RRMS and healthy donors with different HLA-DRB1 genotypes have the potential to efficiently engraft B2m-NOG mice with human T and B lymphocytes and to induce subclinical CNS inflammation dominated by hCD8$^+$ T cells in the brain and the spinal cord. Specifically, mice engrafted with PBMC from DR15-positive MS and healthy donors, and not from a DR13-positive MS donor, showed spontaneous brain and spinal cord infiltration by hCD8$^+$ T cells, and increased infiltration by hCD4$^+$ following immunization for EAE. CNS inflammation was more severe in mice engrafted with PBMC from several DR15-positive MS donors. These mice uniquely developed large human T cell lesions in brain regions including the corpus callosum with or without immunization with myelin peptides using an EAE protocol. Human T cell lesions were associated with locally activated mouse microglia and astrocytes, but notably lacked infiltrating human or mouse monocytes and demyelination, and therefore did not lead to clinical symptoms of EAE or other neurological defects.

PBMC humanized B2m-NOG mice developed several hallmark features of MS that are not readily reproduced in conventional animal models such as EAE. First, brain immune infiltrates in DR15 MS engrafted mice were enriched for hCD8$^+$ over hCD4$^+$ T cells, even though the B2m-NOG model is characterized by increased engraftment of CD4$^+$ over CD8$^+$ T cells (*King et al., 2009a*; *Figure 1B*). The preferential infiltration of the CNS parenchyma by CD8$^+$ T cells and the presence of inflammatory lesions in the brain white and gray matter in our model are both features of MS (*Lassmann, 2019*; *Babbe et al., 2000*; *Wagner et al., 2020*). Second, T cell lesions and locally activated microglia and astrocytes, were observed in brain white matter regions, particularly in the corpus callosum and optic tracts. Lesions in the corpus callosum are important findings in MRI of MS patients and are correlated with cognitive problems as well as disability in the upper limbs (*Ozturk et al., 2010*). Inflammation in the optic tracts is often a first presenting symptom in MS and related disorders (*Jakimovski et al., 2024*). Third, human leukocytes were concentrated on the meninges and choroid plexus, notably in the interventricular foramen connecting the lateral and third ventricles. The choroid plexus is an important immunological site for immune surveillance and a site of CD8$^+$ T cell and granulocyte involvement in progressive MS (*Rodríguez-Lorenzo et al., 2020*).

Nevertheless, MS is a complex disease and immunopathology was only partially reproduced in the brain and the spinal cord of PBMC humanized B2m-NOG mice. Parenchymal lesions contained both hCD4$^+$ and hCD8$^+$ T cells associated with locally activated microglia and astrocytes, but there was poor engraftment by human monocytes in the blood, spleen or CNS of both immunized and non-immunized B2m-NOG mice. Blood myeloid cells are short-lived and constantly renewed by the bone marrow, so that human myeloid cells are nearly absent in the commonly used PBMC-NSG model (*King et al., 2009a*). In contrast, mouse CD45$^+$ cells comprising very high proportions of CD11b$^+$ myeloid cells, Ly6C$^{hi}$ monocytes and Ly6G$^+$ cells, were present in the blood of immunized and non-immunized NOD-*scid* mice, which are a parental strain for B2m-NOG mice, and the non-immunized B2m-NOG mice prior to engraftment. Nevertheless, mouse CD45$^+$ myeloid cells did not infiltrate the CNS of the humanized mice, even after EAE induction. The absence of CNS-infiltrating human or mouse monocytes in the PBMC humanized B2m-NOG mice NOG mice is most likely to be responsible for the absence of clinical symptoms and demyelination in this system. Our results are consistent with a previous study in which NSG mice were humanized with PBMC from healthy donors and immunized

for EAE (*Zayoud et al., 2013*). In that study, mice developed CNS inflammation with infiltrating CD4[+] and CD8[+] T cells but did not show clinical symptoms or demyelination (*Zayoud et al., 2013*). CNS-infiltrating Ly6C[hi]CCR2[+] monocytes, which differentiate into inflammatory macrophages locally in the CNS after being activated by T cells, are critical effector cells in EAE (*King et al., 2009b*; *Ajami et al., 2011*). These cells are massively recruited into the periphery from the bone marrow following immunization (*Dagkonaki et al., 2022*), and are likely to compete for space with engrafted human immune cells before and after EAE induction. The use of immunodeficient NOG/NSG models expressing human transgenes that would support engraftment of human monocytes, or facilitate communication of human T cells with mouse inflammatory monocytes (*Codarri et al., 2013*; *El-Behi et al., 2011*), might allow a more comprehensive modeling of MS immunopathology in humanized mice.

Another limitation for MS modeling in the B2m-NOG mice used in this study is the poor LN structure and functioning that results from depletion of the common cytokine receptor γ-chain gene (*Mebius, 2003*). This deficiency was confirmed in this study by the absence of organized LN and germinal follicles, and of anti-MOG IgG antibodies in the PBMC humanized B2m-NOG mice immunized for EAE. Again, further development of available immunodeficient mouse strains to promote LN organogenesis will be an important step for modeling MS immunopathology in mice. In a previous study, NSG mice injected directly in the CNS with CSF cells from MS patients (*Saeki et al., 1992*) developed clinical symptoms, immune infiltration of the CNS and demyelination, although differences in the route of administration of the transplanted cells might explain the differences between the results of that study and those presented here and by others (*Zayoud et al., 2013*).

Our observation that mice engrafted with PBMC from DR15-positive active RRMS patients showed both spontaneous and EAE-inducible T cell lesions in the brain, brainstem and spinal cord is highly relevant in the context of previous data concerning this haplotype. HLA-DR2 has long been associated with MS (*Nielsen et al., 2009*), with the HLA-DRB1*15 haplotype being linked to earlier disease onset (*Sawcer et al., 2011*; *Masterman et al., 2000*). MHCII molecules present antigens to CD4[+] T cells, and HLA-DRB1*15 is strongly associated with the autoproliferation of peripheral CD4[+] T cells in MS patients (*Mohme et al., 2013*). T cell autoproliferation is mediated by memory B cells in a HLA-DR-dependent manner, and autoproliferating T cells are increased in MS patients treated with natalizumab and enriched for brain-homing T cells (*Jelcic et al., 2018*). In a recent study with humanized NSG mice, CD34[+] hematopoietic progenitor cells (HSC) from HLA-DRB1*15-positive donors induced higher peripheral T cell responses and alloreactivity compared to HSC from HLA-DRB1*15-negative donors, and increased CD8[+] T cell responses after EBV infection (*Zdimerova et al., 2021*). In our study human T cells isolated from DR13 MS mice showed positive T cell proliferation responses to myelin peptides and hCD3 antibody. This provides proof-of-principle for successful engraftment of functional human T cells in humanized mice. However, none of the T cells isolated from DR15 MS1-5 mice showed further proliferation ex vivo, either to myelin peptides or to polyclonal stimuli hCD3 antibody and PHA, possibly due to inherent high levels of autoproliferation in these cells (*Jelcic et al., 2018*). Alternatively, CD4[+] T cells may become nonresponsive to anti-CD3 antibody once engrafted into B2m-NOG mice, as previously reported in NOD-scid mice (*Wagar et al., 2000*). Noteworthy is that MS patients suspected of recent or ongoing reactivation of EBV (DR13 MS, DR15 MS3, and DR15 MS5; *Table 1*), showed the best engraftment of hCD19[+] B lymphocytes in mice, and that both DR15 MS3 and DR15 MS5 mice showed multiple CNS lesions.

There is now strong evidence that EBV infection plays a major role in MS pathogenesis (*Bjornevik et al., 2022*), acting synergistically with HLA-DRB1*15 to increase the likelihood of developing MS (*Nielsen et al., 2009*). HLA-DRB1*15 individuals with specific antibody reactivity to EBNA1 (amino acid 385–420) have 24-fold increased risk for MS (*Sundström et al., 2009*). Anti-EBNA-1 antibodies present high homology and react with several CNS antigens, such as αB-crystallin, MBP, anoctamin 2, and more recently glial cell adhesion molecule (*Lanz et al., 2022*; *Tengvall et al., 2019*; *Hecker et al., 2016*; *Jog et al., 2020*). Nevertheless, positive correlations between the HLA-DRB1*15 genotype, T cell autoproliferation, EBV, and the development of brain lesions in humanized mice remain to be formally tested in a larger study using greater numbers of HLA-DRB1*15-positive MS donors compared to HLA-DRB1*15 negative MS and HLA-DRB1*15-positive healthy controls, together with the measurement of basal levels of T cell proliferation in the peripheral blood of the PBMC donors.

In summary, PBMC humanized B2m-NOG mice show partial representation of MS immunopathology that is not reproduced in conventional animal models such as EAE, particularly the development of

hCD8[+] lesions in the gray and white matter of the brain and spinal cord. New immunodeficient mouse strains developed to support human monocyte engraftment and the development of organized LN therefore hold great promise for comprehensive modeling of MS immunopathology. An important outcome of this study is that PBMC humanized B2m-NOG mice highlight the variability between the different human PBMC donors, and therefore represent a simple and rapid approach for generating immune models for drug screening in vivo at a personalized level.

## Materials and methods
### MS patients and healthy subjects
Blood donors were selected from a Hellenic MS patient cohort attending the MS outpatient clinic at Aeginition University Hospital of the National and Kapodistrian University of Athens (NKUA) School of Medicine, supervised by Dr. Maria Anagnostouli and genotyped for the HLA-DRB1 allele, at the Research Immunogenetics Laboratory, of the same Hospital, as previously described (*Dagkonaki et al., 2020*; *Table 1*). Six patients with long-term RRMS recently presenting with highly active disease since the last dose of immunomodulatory therapy (natalizumab) were selected. Natalizumab treatment was chosen as a criterion for this study because it blocks lymphocyte migration to the brain, thereby increasing circulating T and B lymphocytes (*Yednock et al., 1992*). One patient was HLA-DRB1*13-positive (DR13 MS), and five were HLA-DRB1*15-positive (DR15 MS). The HLA-DRB1*13-positive patient additionally received cortisone treatment 1 month prior to sampling because of high disease activity. A healthy HLA-DRB1*15-matched healthy individual (DR15 HI) was selected as control. Donor characteristics, including viral infections and EBV status are listed in *Table 1*. Blood samples were obtained from MS patients and a healthy individual under signed informed consent in accordance with the Declaration of Helsinki and approval from the Institutional Ethics Committee of Aeginition Hospital, NKUA (Protocol No: 7BSH46Y8N2-B66, 13/05/2015).

### PBMC isolation
Fresh peripheral blood (50 ml) was collected in ethylenediaminetetraacetic acid (EDTA)-treated polypropylene tubes and PBMC were isolated under Ficoll-Histopaque-1077 gradient centrifugation (Sigma-Aldrich). Buffy coats containing blood leukocytes were washed with 2% fetal bovine serum (FBS) in phosphate-buffered saline (PBS). Erythrocytes were removed using Quicklysis erythrocyte lysis buffer (Cytognos) and leukocytes were washed with 2% FBS in PBS. Before transfer of PBMC into mice, we compared several protocols for preparation of human PBMC using blood from a healthy individual, specifically, (1) PBMC isolated from fresh blood on the day of transfer, (2) PBMC isolated from blood after 3 days' storage at 4°C, (3) PBMC cultured for 24 hr in complete RPMI-1640 after snap freezing of fresh PBMC and thawing. Cell viability and analysis of immune cell populations by flow cytometry showed that freshly isolated PBMC gave the best results and were used for transfer in this study (*Figure 1—figure supplement 2*, protocol 1).

### EBV antibody responses in human blood
Blood plasma from the DR13 MS, DR15 MS, and DR15 HI donors was analyzed for EBV antibody responses using the following enzyme-linked immunosorbent assay (ELISA) tests: ELISA-VIDITESTS anti-VCA EBV IgG (ODZ-265), anti-VCA IgM (ODZ-005), anti-VCA IgA (ODZ-096), anti-EA (D) EBV IgG (ODZ-006), anti-EBNA-1 EBV IgM (ODZ-002), and anti-EBNA-1 EBV IgG (ODZ-001) (Vidia). Standardized criteria for the clinical interpretation of EBV antibody responses are shown in *Figure 1—figure supplement 1C*.

### Mice
Ten- to twelve-week-old female immunodeficient B2m-NOD/Shi - *scid IL2rg*[null] (B2m-NOG) mice (stock number 14,957 F: NOD.Cg-B2m<em1Tac>Prkdc < scid>Il2rg <tm1Sug>JicTac; Taconic Biosciences) were used for transfer of freshly isolated human PBMC. These mice lack mature T, B, and NK cells and lack MHC class I molecules. Previous studies show that an equivalent mouse strain, *NOD.Cg-Prkdc*[sci-d]*Il2rg*[tm1Wjl]*B2m*[tm1Unc]*/Sz* (NSG β2m[null]; The Jackson Laboratory) facilitates the engraftment of CD4[+] over CD8[+] T cells and exhibits delayed onset of GVHD (*Morillon et al., 2020a*; *King et al., 2009a*). NOD.

CB17-Prkdc scid/NCrHsd (Envigo) mice were used for investigation of peripheral mouse myeloid cell populations in a related severely immunodeficient mouse strain by flow cytometry.

Mice were transplanted via tail vein injection with $10 \times 10^6$ PBMC in 200 µl HBSS on the same day as cell isolation from fresh peripheral human blood. Mice were assessed daily for symptoms of GVHD, such as fur ruffling, hunched posture, reduced mobility, and tachypnea (*Cooke et al., 1996*), as well as for any spontaneous neurological deficits (*Guyenet et al., 2010*).

All experiments were performed under sterile conditions in the Department of Animal Models for Biomedical Research of the Hellenic Pasteur Institute. Mice were housed in a specific pathogen-free facility in microisolator cages and given ad libitum UV-sterilized standard chow, acidified water and gels for hydration. The experiments complied with ARRIVE guidelines and were in accordance with the local Ethical Committee guidelines on the use of experimental animals at the Hellenic Pasteur Institute, were approved by the national authorities and complied to EU Directive 2010/63/EU for animal experiments. The animal study was reviewed and approved by Committee for Evaluation of Experimental Procedures, Department of Experimental Animal Models, Hellenic Pasteur Institute (Presided by Dr P Andriopoulos pandriopoulos@patt.gov.gr for the Hellenic Republic, General Secretariat for Agricultural Economy, Veterinary and Licenses), and performed under licence numbers 770851/27-11-2019 and 1343917/02-11-2023.

### Immunization with myelin peptides

Peptides MBP83-99, MOG1-20, and mMOG35-55 and hMOG35-55 (S42 in mMOG, P42 in hMOG) were synthesized as previously described (*Tapeinou et al., 2015*). In a previous study, we found that humanized HLA-DR2b transgenic mice lacking all mouse MHCII genes required greater amounts of peptide antigen to induce clinical EAE than wild-type B6 mice (*Dagkonaki et al., 2020*). Due to limited numbers of B2m-NOG mice available for this study, several EAE protocols were first tested using wild-type B6 mice, mainly to exclude the possibility of toxicity at high peptide doses. Briefly, 8-week-old female B6 mice were immunized by subcutaneous (s.c.) tail-base injection of 37 µg (our standard EAE protocol) or 200 µg mMOG35-55, or a myelin peptide antigen cocktail containing 200 µg each of mMOG35-55, hMOG35-55, MOG1-20, and MBP83-99, dissolved in 100 µl saline and emulsified in an equal volume of Freund's complete adjuvant (FCA) (Sigma-Aldrich). FCA was supplemented with 400 µg/injection of H37Ra *Mycobacterium tuberculosis* (Difco, BD Biosciences). All mice, except those immunized using the standard protocol, received an identical boost immunization 7 days later. The immunized mice also received intraperitoneal (i.p.) injections of 200 ng of *Bordetella pertussis* toxin (PTx) (Sigma-Aldrich) at the time of each immunization and 48 hr later. All mice developed clinical symptoms of EAE with no evidence of increased morbidity or mortality at high peptide doses (*Figure 1—figure supplement 3A, B*). For this reason, the high-dose peptide cocktail was chosen for immunization of groups of 12- to 14-week-old PBMC humanized B2m-NOG mice on dpt 14 in EAE experiment 1 (*Figure 1—figure supplement 3C*). In EAE experiment 2, groups of 12- to 14-week-old PBMC humanized B2m-NOG were immunized once, using 100 µg each peptide/mouse, to reduce the possibility of immune tolerance (*Figure 1—figure supplement 3C*). Mice were monitored daily for the clinical symptoms of EAE according to criteria commonly used for scoring EAE: 0, normal; 1, limp tail; 2, hind limb weakness; 3, hind limb paralysis; 4, forelimb paralysis; 5, moribund or dead (0.5 gradations represent intermediate scores) (*Dagkonaki et al., 2020*). Mice were also monitored for signs of other neurological deficits (*Guyenet et al., 2010*).

### MOG antibody responses in PBMC humanized mouse blood

Peripheral blood was collected from immunized and non-immunized mice at sacrifice on dpt 42, and sera collected for analysis of antibody responses. Samples were screened for IgG antibodies against MOG, using indirect immunofluorescence assays (IFA). Sera were analyzed at a starting dilution of 1:10, on EU 90 cells transfected with MOG protein and on EU 90 control transfected cells, according to the manufacturer's instructions (Euroimmun, Lubeck, Germany). Visualization and evaluation of IFA results were performed under a fluorescence microscope (Zeiss, Axioskop 40).

### Fluorescence-activated cell sorting (FACS)

Fresh peripheral blood samples from the MS and HI donors were analyzed by FACS using a panel of immune cell marker antibodies. To monitor immune cell populations, 3 ml fresh peripheral blood

was collected in EDTA tubes and a 100-µl aliquot was used for flow cytometry. Erythrocyte lysis was achieved with with Quicklysis (Cytognos) for 25 min. Cells were washed and incubated with an antibody mixture for human immune cell markers BV-510-hCD45 (clone 2D1), APC-Cyanine 7-hCD3 (clone HIT3a), PE-Cyanine 7-hCD4 (clone A161A1), PerCP-hCD8 (clone SK1), PE-hCD19 (clone 4G7), FITC-hCD56 (clone 5.1H11), APC-hCD66b (clone G1OF5), and BV-421-hCD14 (clone HCD14) (all from Biolegend), for 30 min at 4°C (*Figure 1—figure supplement 1A*). To monitor engraftment of human immune cells in B2m-NOG mice, peripheral blood was collected in heparin tubes from the tail vein at dpt 7, 13, and 42. Erythrocyte lysis and incubation with antibody mixture were performed as for fresh human blood above, except that the APC-hCD66b antibody was replaced by mouse APC-CD45 (clone 30-F11) (Biolegend) (*Figure 1—figure supplement 1A*). Data were acquired with FACSCelesta, FACSCanto II or FACSMelody cytometer and analyzed with FACSDiva (BD) and FlowJo software (Tree Star, Inc).

For intracellular staining of IFN-γ, splenocytes were isolated at sacrifice at dpt 42 as described above, and cells were stimulated with phorbol 12-myristate 13 acetate (PMA) (20 ng/ml, Sigma-Aldrich) and ionomycin (1 µg/ml, Sigma-Aldrich) and treated with brefeldin-A (5 µg/ml, Sigma-Aldrich) for 3 hr at 37°C/5% $CO_2$ in order to disrupt Golgi-mediated transport. Cells were fixed with 2% paraformaldehyde (PFA) in PBS, permeabilized using 0.5% wt/vol saponin, and stained with PerCP-Cyanine 5.5-CD3 (clone HIT3a, Biolegend), antibodies for APC-CD4 (clone RPA-T4, BD Biosciences), PerCP-CD8 (clone SK1, Biolegend), PE-IFN-γ (clone B27; BD Biosciences), and PE-IL-17A (clone eBio64DEC17, eBioscience). Data acquisition was performed using a FACSCalibur cytometer and analyzed with FlowJo software (Tree Star, Inc).

## T cell proliferation assay

Spleens were isolated at sacrifice and cells mechanically separated in RPMI (Invitrogen Life Technologies) containing 10% heat-inactivated FBS. To obtain single-cell suspensions, the washed homogenate was passed through a 70-µm cell strainer and erythrocyte lysis was performed with Gey's erythrocyte lysis buffer for 5 min. The reaction was stopped with RPMI 1640/FBS. Washed splenocytes at a concentration of $10^7$ cells/ml in PBS were incubated with 5 µM carboxylfluorescein succinimidyl ester (CFSE, V12883, Thermofisher) for 15 min at 37°C. Cells were washed and PBS/2% FBS was applied for 30 min at 37°C to stop the reaction. Cells were washed and resuspended in RPMI 1640/FBS with 50 µM 2-mercaptoethanol (Sigma-Aldrich) at concentration of $10^6$ cells/ml. The cells were stimulated in triplicates in round-bottom 96-well plates with a myelin peptide antigen cocktail containing 30 µg/ml each mMOG35-55, hMOG35-55, MOG1-20, and MBP83-99 for 120 hr. Negative control cells were incubated with medium only, and positive control cells were incubated in plates coated with anti-CD3 (1 µg/ml) (clone HIT3a, BD Biosciences) or PHA (2 µg/ml) (Sigma-Aldrich). Results were expressed as cell division index which is the ratio of %CFSE^low splenocytes (proliferating splenocytes) cultured with peptides or anti-CD3 to the of %CFSE^low splenocytes cultured with medium only. Data acquisition was performed using a FACSCalibur cytometer and analyzed with FlowJo software (Tree Star, Inc).

## Histopathology and immunohistochemistry

Mice were transcardially perfused with ice-cold PBS at sacrifice on dpt 42 by carbon dioxide inhalation. Brains, spinal cords, and peripheral tissues (including lung, liver, and inguinal fat tissue) were dissected and post-fixed in 4% PFA fixative overnight at 4°C. Tissues were embedded in paraffin and 5 µm sections used for immunohistochemistry and histopathology. Inflammation in peripheral tissues was visualized by H&E using standard techniques. Immune cell infiltration and inflammation of brain and spinal cord were visualized by immunohistochemistry using antibodies specific for hCD3 (SP7; Epredia), hCD45 (HI30; Biolegend), hCD4 (4B12; Agilent-Dako), hCD8 (C8/144B; Agilent-Dako), mouse CD45 (30-F11; Biolegend), mouse Iba1 (019-19741; Wako chemicals), and mouse GFAP (Z0334; Agilent-Dako). Primary antibodies to the human immune cell markers were visualized using an EnVision FLEX High pH kit (Agilent-Dako). Antibodies to mouse markers were detected using biotinylated secondary anti-IgG antibodies followed by horseradish peroxidase-labeled avidin–biotin complex, and signal development with 3'3'-diaminobenzidine (all Vector Laboratories). Nuclei were counterstained with hematoxylin. Images were captured with an Olympus DP71 microscope digital camera using cell^A imaging software (Soft Imaging System GmbH). For measurements of brain areas Image J software was used. Semi-quantitative scoring of hCD3-positive T cells in whole spinal cord

sections was perfomed as follows: 0, 0-3 cells; 0.5, 3-10 cells; 1, 10-25 cells; 2, 25-50 cells; 3, 50-90 cells; 4, more tha 90 cells. High-resolution immunofluorescence imaging was performed by confocal microscopy using paraffin sections immunostained with marker antibodies for myelin (rat anti-MBP; ab7349; Abcam), microglia (rabbit anti-Iba1; 019-19741; Wako chemicals) followed by anti-rat CF-647 (Biotium) and anti-rabbit AlexaFluor 568- (A11011; Invitrogen) labeled secondary antibodies, respectively. Nuclei were counterstained with DAPI (D1306; Invitrogen). A Leica TCS SP8 confocal microscope was used to acquire fluorescent images.

## Statistical analysis

Data were processed using Microsoft Excel, and statistical analysis was performed with GraphPad Prism 8. Figures were made using Adobe Illustrator V24.3. Data were analyzed with Student's $t$ test and one-way analysis of variance and results are presented as means ± standard error of the mean. Results were considered statistically significant when $p \leq 0.05$.

## Acknowledgements

We wish to thank Ivan Gladwyn-Ng and Dimitri Gimnopoulos (Taconic Biosciences) for providing B2m-NOG mice and expert scientific support, Konstantinos Kambas (Molecular Genetics Lab, HPI), Maritsa Margaroni (Flow Cytometry Unit, HPI), Katerina Nanou and George Kollias (BSRC Alexander Fleming) for support with the flow cytometry analyses, Maria Belimezi and Manolis Angelakis (Diagnosis Dept, HPI) for screening mouse plasma for anti-hMOG IgG antibodies, and Maria Avloniti for contributing research material. We especially thank Eirini Fragkiadaki and Ariadne Karles, and other members of the Department of Experimental Animals for Biomedical Research of HPI for their help and expert support in maintenance of the B2m-NOG mice, and the training of experimenters in mouse familiarization techniques during these experiments. This research was co-financed by European Union and Greek national funds by The Management and Implementation Authority for Research, Technological Development and Innovation Actions (MIA-RTDI/ΕΥΔΕ-ΕΤΑΚ) of the Hellenic Ministry of Development and Investment, through the Operational Program Competitiveness, Entrepreneurship and Innovation, under the call RESEARCH – CREATE – INNOVATE (project code T1EDK-01859; acronym AKESO) to LP. Research was also supported by the Hellenic Ministry of Development and Investment, General Secretariat for Research and Innovation (GSRI) Flagship Action for Neurodegenerative Diseases on the basis of Personalized Medicine (Project code 2018ΣΕ01300001, acronym EDIA-N) to LP. VG was supported by the Hellenic Pasteur Institute through a 'NOSTOS FOUNDATION TRUST FUND' PhD fellowship.

## Additional information

### Funding

| Funder | Grant reference number | Author |
| --- | --- | --- |
| Hellenic Ministry of Development and Investment | T1EDK-01859 | Lesley Probert |
| Hellenic Ministry of Development and Investment | 2018ΣΕ01300001 | Lesley Probert |

The funders had no role in study design, data collection, and interpretation, or the decision to submit the work for publication.

### Author contributions

Irini Papazian, Data curation, Formal analysis, Investigation, Methodology, Writing – review and editing, Designed, set-up and performed experimental protocols for PBMC preparation and humanization of B2m-NOG mice; Monitored mice for clinical symptoms and performed immunohistochemical analysis of CNS tissues; Analysed results and prepared figures; Maria Kourouvani, Investigation, Writing – review and editing, Performed quantitative immunohistochemical analysis of CNS tissues; Analysed

results; Monitored EAE mice for clinical symptoms and prepared figures; Anastasia Dagkonaki, Investigation, Methodology, Writing – review and editing, Designed and performed EAE experiments; Monitored mice for clinical symptoms; Performed immunohistochemistry of LN tissues and FACS analyses of humanized mice; Analysed results and prepared figures; Vasileios Gouzouasis, Investigation, Writing – review and editing, Performed FACS analysis and EBV ELISA on fresh human donor blood samples; Lila Dimitrakopoulou, Investigation, Methodology, Writing – review and editing, Designed human FACS antibody panel; Performed analyses of human and mouse blood and analysed results; Nikolaos Markoglou, Resources, Data curation, Writing – review and editing, Collected and co-ordinated delivery of human donor blood samples; Fotis Badounas, Investigation; Theodore Tselios, Resources, Designed, synthesized, purified and characterized myelin peptide analogues; Maria Anagnostouli, Resources, Data curation, Supervision, Writing – review and editing, Performed HLA genotyping; Selected patients for this study; Supervised Hellenic MS patient cohort; Lesley Probert, Conceptualization, Formal analysis, Supervision, Funding acquisition, Methodology, Writing – original draft, Project administration, Writing – review and editing, Responsible for overall experimental design and management of project; Analysed and interpreted data; Wrote paper

### Author ORCIDs
Lesley Probert ⓘ https://orcid.org/0000-0002-5931-2880

### Ethics
The animal study was reviewed and approved by Committee for Evaluation of Experimental Procedures, Department of Experimental Animal Models, Hellenic Pasteur Institute (Presided by Dr P Andriopoulos pandriopoulos@patt.gov.gr for the Hellenic Republic, General Secretariat for Agricultural Economy, Veterinary and Licenses). Licence numbers 770851/27-11-2019 and 1343917/02-11-2023.

Joint Public Review: https://doi.org/10.7554/eLife.88826.3.sa1
Author response https://doi.org/10.7554/eLife.88826.3.sa2

---

## Additional files

### Supplementary files
• MDAR checklist

### Data availability
All data generated or analyzed during this study are included in the manuscript and supporting files.

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
