## [Editor Report · eLife assessment]

The humanized immune system model represents a **valuable** model in which to evaluate mechanisms that may drive MS processes in vivo. The data are **solid** given the revisions and expansion of numbers of mice to yield more statistical rigor. This model will be used by the greater community studying MS pathophysiology.

---

## [Referee Report · Joint Public Review]

The premise of this work carries great potential. Namely, developing a humanized mouse system in which features of adaptive immunity that contribute to inflammatory demyelination can be interrogated will allow for traction into therapeutics currently unavailable to the field. Immediate questions stemming from the current study include the potential effect of ex vivo activation of PBMCs (or individual T and B cells) in vitro prior to transfer as well as the TCR and BCR repertoire of CNS vs peripheral lymphocytes before and after immunization. This group has been thoughtful and clever about their approach (e.g. use of subjects treated with natalizumab), which gives hope that fundamental aspects of pathogenesis will be uncovered by this form of modeling MS disease.

Multiple sclerosis is an inflammatory and demyelinating disease of the central nervous system where immune cells play an important role in disease pathobiology. Increased incidence of disease in individuals carrying certain HLA class-II genes plus studies in animal models suggests that HLA-DRB1*15 restricted CD4 T cells might be responsible for disease initiation, and other immune cells such as B cells, CD8 T cells, monocytes/macrophages, and dendritic cells (DC) also contribute to disease pathology. However, a direct role of human immune cells in disease is lacking to a lag between immune activation and the first sign of clinical disease. Therefore, there is an emphasis on understanding whether immune cells from HLA-DR15+ MS patients differ from HLA-DR15+ healthy controls in their phenotype and pro-inflammatory capacity. To overcome this, authors have used severely immunodeficient B2m-NOG mice that lack B, T cells and NK cells and have defective innate immune responses and engrafted PBMCs from 3 human donors (HLA-DR15+ MS and HI donors, HLA-DR13+ MS donor) in these B2m-NOG mice to determine whether they can induce CNS inflammation and demyelination like MS.

The study's strength is the use of PBMCs from HLADRB1-typed MS subjects and healthy control, the use of NOG mice, the characterization of immune subsets (revealing some interesting observations), CNS pathology etc. Weaknesses are lack of phenotype in mice and no disease phenotype even in humanized mice immunized for disease using standard disease induction protocol employed in an animal model of MS, and lack of mechanistic data on why CD8 T cells are more enriched than CD4+ T cells. The last point is important as postmortem human MS patients' brain tissue had been shown to have more CD8+ T cells than CD4+ T cells.

Thus, this work is an important step in the right direction as previous humanized studies have not used HLA-DRB1 typed PBMCs however the weaknesses as highlighted above are limitations in the model.

---

## [Author Response]

The following is the authors’ response to the original reviews.

We provide below a point-by-point reply to the Reviewers, and hope that our new manuscript will now meet the Reviewers’ concerns and the requirements for publication in eLife.

In summary, we have performed a new set of mouse humanization experiments using a new cohort of 4 additional HLA-DRB1*15-typed MS patients as donors, all presenting with highly active disease and under treatment with natalizumab. The new experiments aim to strengthen and further extend the findings of the original paper that HLA restriction rather than disease status plays an important role in the development of CNS inflammation. Additionally, we performed EAE using a revised protocol using lower amounts of peptide antigens to reduce the possibility of immune tolerance. Indeed, our original observations were further enriched with the finding that immunization increases infiltration of the CNS by human CD4 T cells, a finding consistent with EAE pathology, and that these human CD4 T cells co-localize with human CD8 T cells in the brain lesions. Further, we provide more detailed information concerning the EBV infection status of the PBMC donors used for humanization and find some first indications of relationships between the B cell engraftment in humanized mice, EBV status of the donors and the development of brain lesions that might stimulate further investigation in future studies.

**Point-by-point reply to reviewers:**

**Reviewer 1:**

We thank Reviewer 1 for their valuable comments, and for their support of the overall approach as a model system. We have addressed the comments by providing additional requested information, as well as performing a EAE with a revised protocol, as suggested. We believe the new results significantly upgrades the information gained from this study.

(1) Throughout their paper, the authors never quantify the difference in CD4 vs CD8 T cell infiltration into the CNS. While repeatedly claiming that there are fewer CD4 T cells present than CD8 T cells within the CNS, this data is not included. Further, spinal cord numbers of CD4 and CD8 are not provided in lieu of CD3 T cell characterization.

Reply: We have now included quantitative data for the differences in CD4 vs CD8 T cells in the brain and spinal cord of non-immunized and EAE immunized mice. Thus, in brain (Fig. 2E) and spinal cord (Fig. 3D) of non-immunized mice, and brain (Fig. 4D, E, L) and spinal cord (Fig. 5D) of immunized mice we show data for numbers of hCD8 and hCD4 T cells, and ratios of CD4 to CD8 in at borders and parenchyma. Notably, using a revised EAE protocol in the second set of experiments, we observed a marked increase in hCD4 T cell infiltration at the CNS borders and parenchyma, an observation consistent with successful EAE immunization.

B cells don't make up any significant component of the cells transferred from HLA-DR15 donors. While the cells transferred from the HLA-DR13 donor are composed of a considerable number of B cells, the mice that received these cells didn't develop any signs of neurologic disease.

In the second experiment using new DR15 MS donors, we observed significant B cell engraftment also in several groups of DR15 MS mice. With the additional groups of mice, we were able to see a relationship between B cell engraftment in DR13 and DR15 MS mice with indicators of recent or ongoing reactivation of EBV. This is an interesting preliminary observation that might be tested in future larger studies.

(2) Incomplete exploration of potential experimental autoimmune encephalomyelitis (EAE) modeling. Comparison of the susceptibility of B2m-NOG mice to EAE dependent on various peptide doses would be highly informative. Given that the number of hCD45+ in the periphery of NOG mice decreases following this immunization it would be prudent for the authors to determine if such a high peptide dose is truly ideal for EAE development in this mouse model.

Reply: We thank the reviewer for this critical comment. In the second group of experiments (DR15 MS2-5), we revised the EAE protocol to use lower amounts of peptides in a single immunization, thereby greatly reducing the exposure of human T cells to antigen and risk of tolerance/anergy. This resulted in (i), by-pass of the reduction in proportions of peripheral hCD45 cells following immunization in the peripheral blood (Fig. 1A), and (ii), increased numbers of hCD4 T cells and hCD4/hCD8 T cell ratios at the borders and infiltrating the parenchyma of brain (Fig. 4D,E) and spinal cord (Fig. 5D).

(3) The degree of myelin injury is not presented. The statement is repeatedly made that "demyelination was not observed in the brain or spinal cord" but no quantification of myelin staining is shown.

Reply: The reviewer refers to a pivotal feature (and limitation) of this particular humanized model. Despite significant T cell infiltration of white and grey matter regions of brain and spinal cord, there is no detectable demyelination. This has also been reported by in independent study using a similar humanized system (Zayoud et al., 2013). We have supplemented the figures with photomicrographs showing the presence of unperturbed myelin in the corpus callosum white T cell lesions (Fig. 4F, inset stained with Luxol fast blue), and a confocal micrograph in the same region double-immunostained for hCD45 immune cells and MBP (Fig. 4G).

Minor points:Method of quantification (e.g. cells per brain slice in figures 2E; 4E) is not very quantitative and should be justified or more appropriately updated to be more rigorous in methodology.

Reply: In the new figures, we have changed the method of quantification of brain parenchyma infiltrating cells from per brain slice, to cells per tissue area mm2 (Fig. 2D, Fig. 4D).

Fig. 4 data should be shown from un-immunized DR15 MS and DR15 HI mice.

Reply: We now include the quantitative data from un-immunized mice compared to immunized mice in all groups (Fig. 4 C-E).

**Reviewer 2:**

We thank Reviewer 2 for their very pertinent comments and overall for highlighting the importance of humanized mice as an approach for further understanding the pathobiology of MS. We also thank this reviewer for their positive comments concerning the study design, specifically the use of fresh PBMC isolated from HLADRB1-typed MS individuals and healthy control. The reviewer highlights 4 major weaknesses of the study that we have tried to address in order to increase the value of the study.

(i) Lack of sufficient sample size (n=1 in each group) to make any conclusion.

Reply: We have increased the sample size for the DR15 MS group from n=1 to n=5 by generating new humanized mice using PBMC freshly isolated from additional MS donors, all HLA-DRB1*5 with active RRMS and under treatment with natalizumab. Here we were able to maximize on our excellent collaboration with neurologists at the neighboring University Hospital, which runs a large organized MS outpatient clinic, with HLADRB1-typed MS individuals that are closely monitored over the course of their disease and therapy. In this way, we were able to address the engraftment success of human immune cells and variability in CNS lesion development across mice generated from 5 different DR15 MS patients. We also monitored markers for EBV activation status in all the patients used for mouse humanization in this study.

(ii) Lack of phenotype in mice.

Reply: As already described in the results and address in the discussion, the B2m-NOG immunodeficient mouse strain used here is a state-of-the-art experimental tool for humanization studies, but unfortunately fails to support engraftment by human monocytes. We and previous groups (Zayoud et al., 2013) show that CNS lesions in humanized mice contain high numbers of hCD4 and CD8 T cells, accompanied by locally activated murine microglia and astrocytes, but lack human monocytes. The humanized mice contain large proportions of immature mouse CD11b+Ly6Chi monocytes in the periphery (Suppl. Table 4) but these cells are not recruited into the CNS in non-immunized or immunized humanized mice, potentially due to incompatible chemokine signals across mouse/human. The absence of human monocyte engraftment in this model is the most likely reason that lesions do not demyelinate and this limitation of the currently available host mouse strains is one that needs to be addressed before full modelling of CNS demyelination by human immune cells can be achieved.

(iii) No disease phenotype even in humanized mice immunized for disease using standard disease induction protocol employed in an animal model of MS.

Reply: As described above, following the suggestion of reviewer 1 (point 2) we revised the EAE protocol to use lower amounts of peptides given as a single immunization. This resulted in increased numbers of hCD4 T cells and the hCD4/hCD8 T cell ratios at the borders and infiltrating the parenchyma of brain (Fig. 1E, Fig. 2D) and spinal cord (Fig. 5D), all indicative of a successful EAE immunization. Although immunized mice showed lesions with mixed populations of hCD4 and hCD8 T cells, demyelination and therefore clinical symptoms were again not observed. As outlined in (ii) above, successful human monocyte engraftment would be fundamental for the development of demyelination and clinical symptoms in PBMC humanized mice, and new immunodeficient animal strains should be developed to achieve this.

(iv) Mechanistic data on why CD8 T cells are more enriched than CD4+ T cells.

Reply: The question of why hCD8 T cells are more enriched in the CNS than hCD4 cells is answered at least in part by the results from our new EAE experiments, which clearly show that immunization increases CNS infiltration by hCD4 T cells versus hCD8 T cells. In general, EAE protocols are designed to activate antigen-specific CD4 T cells and this is verified in the CNS of immunized humanized mice, where hCD4 T cells infiltrate to join hCD8T cells in lesion areas. The predilection of hCD8 T cells for CNS is obvious in non-immunized humanized mice, especially in the parenchyma (see Fig. 2E) and MS patients, while hCD4 infiltration becomes important after EAE immunization. The humanized model system might therefore represent a unique tool for studying mechanisms underlying preferential hCD8 T cell involvement in MS neuroinflammaton, a system that is not accurately modelled in current EAE models. As this reviewer correctly points out, this is very important point as postmortem MS patients’ brains have more CD8 T cells than CD4 T cells.